# Composite selection signal analysis: Uncovering candidate genes and quantitative trait loci in Indian sheep breeds

**Sapna Nath**[1☯‡], **Satish Kumar Illa**[2☯‡], **Destaw Worku**[3*], **Sabyasachi Mukherjee**[4],
**Anupama Mukherjee**[4], **Vinod Kumar Yata**[5]

**1** College of Veterinary Science, Garividi, Sri Venkateswara Veterinary University, Andhra Pradesh,
India, **2** Livestock Research Station, Garividi, Sri Venkateswara Veterinary University, Andhra Pradesh,
India, **3** Department of Animal Sciences, College of Agriculture and Environmental Sciences, Bahir Dar
University, Bahir Dar, Ethiopia, **4** Division of Animal Genetics and Breeding, Indian Council of Agricultural
Research-NationalDairy Research Institute, Karnal, Haryana, India, **5** Department of Pharmacology,
School of Allied and Healthcare Sciences, Malla Reddy University, Hyderabad, Telangana, India

☯ These authors contributed equally to this work.
‡ These authors share first authorship on this work.
* destawworku@gmail.com

org/10.1371/journal.pone.0344299

University Faculty of Agriculture, EGYPT

**Peer Review History:** PLOS recognizes the
benefits of transparency in the peer review
process; therefore, we enable the publication
of all of the content of peer review and
author responses alongside final, published
articles. The editorial history of this article is
available here: https://doi.org/10.1371/journal.
pone.0344299

## Abstract

Selective pressures, both natural and artificial, have significantly influenced the
genomic architecture of domesticated sheep. Understanding their underlying molec-
ular mechanisms is critical for developing efficient breeding programs to conserve
and improve economically important traits in native breeds. In this study, we anal-
ysed high-density 50K SNP data from three Indigenous sheep breeds: Chanthangi
(CHA, n = 29), Garole (GAR, n = 24), and Deccani (IDC, n = 26), each native to diverse
climatic regions of India. We implemented a novel SNP-based de-correlated compos-
ite of multiple signals (DCMS) statistic, which integrates *p*-values from five selection
metrics viz., FST, H1, H12, Tajima's D, and nucleotide diversity (π) into a unified
measure. The SNP-based DCMS approach offers finer resolution and complements
window-based methods by enabling more precise localisation of selection signals
and candidate genes. Multiple testing correction was applied at a False Discovery
Rate (FDR) threshold of <5% to detect significant genomic regions. Comprehensive
gene and quantitative trait loci (QTL) annotation and enrichment analysis of these
regions were also performed for each breed. The DCMS analysis identified 21, 10,
and 14 novel and breed-specific putative genes in the Chanthangi, Garole, and
Deccani breeds, respectively, as well as 10, 28, and 13breed-specific QTL regions.
The identified genes and QTLs are associated with diverse phenotypic traits, includ-
ing growth and muscle development (*CNTNAP5, DOCK3*), reproduction (*TCERG1L,
BUB1, UNC5C, C2CD5, BBX*), wool trait (*TPPP3, P2RY6, FGF10, POU2F1,
FAM168A*), disease resistance (*MTSS1, B4GALNT3*), environment adaptation
(*TRMT12, MAPKAPK3*), domestication (*LRRC36*). The QTLs identified are associ-
ated with body conformation (body measurements and bone area), production (milk

**Data availability statement:** The genotypic data of indigenous Sheep breeds of India (Chantangi, Garole, and Deccani; n=29, 24, and 26, respectively) were obtained from the WIDDE database with the following link: http://widde.toulouse.inra.fr/widde/widde/main.do?module=sheep.

**Funding:** The author(s) received no specific funding for this work.

**Competing interests:** The authors have declared that no competing interests exist.

fat yield), reproduction (total lambs born), disease resistance (hemonchus resistance, foot rot, and pneumonia susceptibility), and health (platelet count and entropion). Our SNP-based DCMS method enabled high-resolution detection of breed-specific selection signatures. It facilitated the discovery of both known and novel genomic regions, candidate genes, and QTLs unique to Indian sheep breeds. This comprehensive approach provides valuable insights into the molecular mechanisms underlying economically important traits and offers a robust foundation for targeted genetic improvement and conservation of indigenous sheep breeds.

## 1. Introduction

Natural selection enhances the fitness of individuals, thereby increasing the frequency of beneficial mutations in a population [1]. Selection pressure leaves distinct patterns on the genome, known as selection signatures [2,3], which provide fundamental insights into the evolutionary forces shaping genetic diversity. Selection signatures also aid in QTL mapping of quantitative traits in domestic animals and in identifying advantageous mutations in livestock populations [4]. SNP genotyping arrays have revolutionized the study of genomic variation and are now a standard tool for detecting selection signatures in populations. These arrays consist of genetically stable single-nucleotide polymorphisms distributed across the genome, facilitating large-scale genetic analyses. Several methods have been developed to identify signatures of selection, including those based on allele frequency spectra, linkage disequilibrium, and population differentiation [5,6]. Such analyses have been applied to various sheep breeds worldwide [7–9]. To increase resolution and accuracy, composite methods, such as the composite of signals (CMS), have been proposed that combine information from multiple statistics to enhance the detection of selection signatures [10]. Methods like the integral haplotype score are effective for detecting incomplete or ongoing selection. In comparison, XPEHH identifies signals that are nearly fixed within one group but variable across the other. Consequently, most studies in farm animals now use complementary methods to improve signal resolution [5,11].

To further enhance accuracy and resolution, genomic regions under selection can be detected with greater accuracy and higher resolution using composite methods. These approaches, such as the decorrelated composite of multiple signals (DCMS), effectively combine signals from multiple single statistics while accounting for their correlations [12,13]. This study used a high-density SNP array, the Illumina OvineSNP50 BeadChip, widely used in sheep genomic research, which includes polymorphic SNPs from both Indian and global sheep breeds, to generate genomic data for the three sheep breeds. Such high-density SNP data is crucial for understanding the genetic landscape of Indian sheep [14].

India boasts a substantial sheep population of 74.44 million, comprising 44 registered breeds, according to the 20th Livestock Census [15]. These breeds exhibit significant genetic diversity and adaptability to a variety of ecological environments, from the cold, dry regions of the Western Himalayas to the hot, dry areas of Rajasthan

and nearby states, the semi-arid Deccan Plateau, and the sub-humid regions of West Bengal and Odisha. Consequently, the observed morphological, physiological, and behavioral variations among these breeds are directly due to differences in the environments in which they were raised, reflecting their specific adaptations. The present study focuses on elucidating the genomic basis of these adaptations in the three Indian sheep breeds: Changthangi, Garole, and Deccani. The Changthangi breed is well adapted to the cold, arid environments of the high-altitude Himalayan regions. The Garole breed, renowned for its high twinning rate, is found in the hot, humid conditions of West Bengal. The Deccani breed is native to the arid regions of the Deccan Plateau and is characterized by a black coat and coarse wool [16].

Genomics can help identify variants associated with productivity, disease resistance, and adaptability to diverse environmental conditions, enabling more informed selection and conservation decisions. Genomic selection programs can optimise traits such as growth rate, wool quality, and reproductive efficiency while preserving the unique characteristics of native sheep. However, despite the rich phenotypic diversity observed in Indian sheep, most genomic studies have focused on either a single statistic or broad, window-based comparative frameworks, often overlooking the fine-scale selection signatures unique to indigenous breeds [17–20].

To address these gaps, the present study applies a novel SNP-based DCMS approach, integrating five univariate statistics (FST, H1, H12, Tajima's D, and nucleotide diversity π) for high-resolution detection of selection signatures in three indigenous Indian sheep breeds: Changthangi, Deccani, and Garole. Unlike previous window-based or cross-population studies, our SNP-level strategy enables precise localization of candidate genes and QTLs under selection, facilitating the identification of both known and novel trait-associated loci. Additionally, we perform comprehensive breed-specific gene and QTL annotation and enrichment analysis, linking genomic regions under selection to economically important traits and providing actionable insights for genetic improvement and conservation of Indian sheep.

## 2. Materials and methods

### 2.1. Ethics statement

The research did not involve human or animal participants, and all data examined were publicly available. Therefore, ethical approval or consent was not necessary.

### 2.2. Animal selection and genotyping

In the present study, the high-density SNP genotypes of three Indian sheep breeds, i.e., Changthangi (CHA, n = 29), Deccani (IDC, n = 24), and Garole (GAR, n = 26), were obtained using the Illumina Ovine SNP50 BeadChip, which was sourced from the WIDDE (Web-Interfaced next generation Database dedicated to genetic Diversity Exploration) [21]. This database is designed to store and manage high-density genotyping datasets for bovine and ovine species. The Illumina Ovine SNP50 BeadChip consists of 54,241 SNPs distributed across the sheep genome, with an average density of one SNP every 51 kb. Quality control (QC) of the genotypes was performed based on the following criteria: minor allele frequency (maf < 0.05), Hardy-Weinberg equilibrium (HWE p < 0.0001), individuals with missing genotypes (--mind < 0.1), and SNPs with missing genotype data (geno < 0.1) [20]. A total of 41,997 SNPs were retained for further analysis. These QC criteria were applied to eliminate low-quality SNPs and to advance to subsequent phases of the analysis. Our study employed stringent SNP-level quality control criteria to ensure robust and accurate variant calls. SNPs with unknown genomic coordinates or located on sex chromosomes were excluded, restricting analyses to well-mapped autosomal loci. The SNPs with precise coordinates based on the OAR (*Ovis aries* Reference) Rambouillet reference genome assembly were included for analysis.

### 2.3. Principal component analysis

Principal component analysis (PCA) was performed on SNP genotypes from the studied sheep breeds to investigate the structure and clustering of individuals in the dataset. PCA categorized individuals into distinct clusters, and

selection signatures were then analyzed within each cluster. The PCA analysis was performed in R using the *snprelate* package [22].

## 2.4. De-correlated composite of multiple selection signals

In this study, we computed the DCMS statistic by combining five univariate statistics: FST, H1, H12, Tajima's D, and nucleotide diversity π [23–27].

The computation of the DCMS statistic for a locus l is given below

$$DCMS_l = \sum_{t=1}^{n} \frac{\log\left[\frac{1-P_{lt}}{P_{lt}}\right]}{\sum_{i=1}^{n} |r_{it}|}$$

(1)

The DCMS statistic combined the *p*-values of individual statistics and their correlations to improve the sensitivity and robustness of detecting selection signals. The DCMS statistic was composed of two components. The numerator component included $P_{lt}$, which indicated the *p*-value at a given locus '*l*' of each univariate statistic '*t*'. The denominator contained the correlation parameter '$r_{it}$', which accounted for the covariance structure among all the included univariate statistics [13].

## 2.5. Univariate statistics in DCMS

**2.5.1. Fixation index ($F_{ST}$).** The $F_{ST}$ index denoted the degree of differentiation between the populations. The functions in *PLINKV1.9*, such as —-fst and —-within, were used to determine the FST values for each SNP, and the negative FST values were changed to zeros. Later, the FST values were smoothed using the *runmed* function in R. The $F_{ST}$ values for each breed were estimated.

**2.5.2. Haplotype homozygosity statistics (H1 and H12).** Haplotype homozygosity assesses the conservation of haplotypes and can help identify regions under selective pressure. The haplotypes in each data set were determined using the SHAPEIT2 program [28]. The conditional states parameter was kept at its default value for haplotype phasing. The *SNePv1.1* program computed the effective population size for each breed, which was then used to phase the chromosomes [29]. A high-resolution Ovine recombination map was used to correct for recombination rate variation across all autosomes [30]. An R script was used to convert the phased haplotypes into the input files required by the H12_H1H2.py program. The window size of SNPs was kept to 14, and the step size was 1 (-window 14 -jump 1) [26].

**2.5.3. Tajima's D and nucleotide diversity (π).** Tajima's D detects departures from neutrality, with negative values indicating potential selective sweeps, while nucleotide diversity (π) measures the genetic variation within a population. The statistics Tajima's D and (π) at each coordinate in the dataset of all the studied breeds were computed using the *vcftools* program [31], and the estimates were computed for each chromosome separately. A 300 MB nonoverlapping sliding window (-Tajima 300) was used to compute Tajima's D index. Furthermore, *p*-values were assigned to SNPs within the bin. Meanwhile, the –site-pi option in *vcftools* is used to compute the π values for each SNP. The *runmed* function smoothed the *p*-values as outlined in [27].

**2.5.4. DCMS estimation.** The DCMS statistic was computed by integrating the *p*-values from five univariate selection statistics (FST, H1, H12, Tajima's D, and nucleotide diversity π) using the stat_to_*p*-value function within the *MINOTAUR* R package, based on fractional ranks [32]. To account for correlations among these individual statistics, a correlation matrix of order n x n was computed using the *covNAMcd* function from the *rrcovNA* R package (alpha = 0.75, nsamp = 50,000) [33]. This correlation matrix served as input for DCMS estimation, and the resulting DCMS values were normalized using the rlm function from the *MASS* R package [34], as described in Yurchenko et al. (2018) [27]. This approach yields a composite signal that accounts for the covariance structure among the individual statistics, thereby

improving the robustness of selection signal detection [13]. To reduce the error rate from multiple tests [35]. The false discovery rate (FDR) was controlled using *q*-values transformed from *p*-values, as described in [36], to identify significant genomic regions [36].

## 2.6. Identification of functional genes and QTL regions

In this genome-wide selection signature analysis, the *p*-value represents the probability of observing a test statistic at least as extreme as the observed value under the null hypothesis of no selection, with smaller values indicating stronger evidence of a selection signal. However, testing thousands of SNPs simultaneously inflates the risk of false positives, which is controlled by applying false discovery rate (FDR) correction to produce *q*-values—the expected proportion of false discoveries among significant results. A *q*-value threshold of 0.05 ensures that, on average, at most 5% of declared significant loci are false positives, providing a reliable, genome-wide-corrected interpretation of the selection signatures [36,37]. The putative genomic regions were identified using the *q*-value. A *q*-value threshold of 0.05 was chosen to identify genomic regions with strong evidence of selection, while adjacent SNPs with *q*-values greater than 0.1 were excluded to reduce false positives. The gene and QTL annotation of the candidate genomic regions was carried out using the Genomic Annotation in Livestock for positional candidate Loci (*GALLO*) package v4.2.0 [38]. The '.gtf' file related to ARS-UI_Ramb_v2.0 of the sheep genome was accessed from NCBI, and the corresponding '.gff' file, which belonged to the ARS-UI_Ramb_v2.0, was downloaded from the Sheep QTL database [39].

Candidate genes and quantitative trait loci (QTLs) located within the significant genomic regions are considered putative if they fall within an interval of 0.5 Mb upstream and downstream. Furthermore, QTL enrichment analysis was performed to identify overrepresented traits within the significant regions, using a genome-wide approach and a false discovery rate (FDR) threshold of 0.05.

## 3. Results

### 3.1. Quality control of genotypic data

The quality control of genotypic data included filtering SNPs with minor allele frequency (maf < 0.05), assessing Hardy-Weinberg equilibrium (hwe < 0.0001), and setting missingness thresholds for SNP genotypes (geno < 0.1) and individual genotypes (mind < 0.1). The SNPs excluded for maf, hwe, geno, and mind were 3845, 299, 678, and 0, respectively. Furthermore, SNPs with unknown coordinates (n = 1232) on chromosomes other than the autosomes were removed. The final dataset contains 41997 variants for downstream analysis.

### 3.2. Principal component analysis (PCA)

Principal component analysis (PCA) was employed to ascertain distinct clusters of SNP genotypes among the three Indian sheep breeds under investigation. The results of PCA are presented in Fig 1. The PCA confirmed the genetic differentiation among the three breeds, as indicated by their clear separation into distinct clusters. PC1 accounted for 14.34% of the variation, highlighting the strongest genetic differences between the populations, while PC2 explained an additional 7.98% of the total variation, capturing finer distinctions within the populations.

### 3.3. De-correlated composite of multiple selection signals (DCMS)

The DCMS analysis identified 45 breed-specific regions under selection, with 21 in Chanthangi, 10 in Garole, and 14 in Deccani sheep genomes. The total size of these regions was 2871.54 Kb for Chanthangi, 2857.84 Kb for Garole, and 2513.30 Kb for Deccani. The average length of these regions was 136.74 ± 229.77 Kb in Chanthangi, 259.80 ± 310.04 Kb in Garole, and 167.55 ± 216.86 Kb in Deccani. The genome lengths covered ranged from 1.83 Kb to 1.08 Mb in Chanthangi, 34.11 Kb to 1080.32 Kb in Garole, and 2.20 Kb to 817.32 Kb in Deccani.

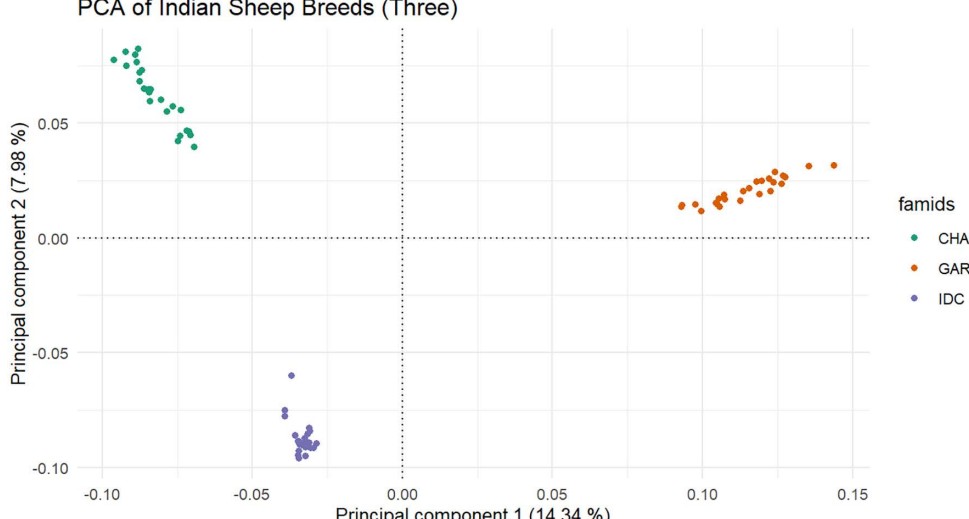

**Fig 1. The PCA analysis showing the population structure of three Indian sheep breeds.** The PCA plot reveals genetic differentiation and clustering among individuals of these breeds. The x-axis, representing Principal Component 1 (PC1), accounts for 14.34% of the total genetic variation and mainly distinguishes Garole from the Changthangi and Deccani breeds. An additional 7.98% of the variation is explained by Principal Component 2 (PC2) on the y-axis, further separating the Changthangi and Deccani breeds into two distinct clusters. This figure clearly demonstrates genetic stratification and unique ancestry in these breeds, reflecting their distinct evolutionary histories and adaptations to different environments.

### 3.3.1. Chanthangi sheep (gene annotation).

In the present study, the gene annotation of the significant genomic regions uncovered twenty-one potential candidate genes related to a variety of traits in this sheep, including *CNTNAP5* (body growth), *MTSS1* (host resistance to bacterial infections), *TRMT12* (immune system and environment adaptation), *TPPP3* (hair follicle growth), *LRRC36* (domestication and early development), *STARD10* (resilience to paratuberculosis), *P2RY6*, *FGF10* and *FAM168A* (hair follicle morphogenesis), *ARHGEF17* and *NIM1K* (meat quality), *GHR* (litter size). These genes were located on OAR 2, 3, 9, 14, 15, and 26. A greater number of significant genomic regions were identified on OAR 15 (Table 1).

### 3.3.2. Quantitative trait loci (QTL) annotation and enrichment.

Table 2 presents the results of QTL annotation in Chanthangi sheep. Our study identified significant genomic regions annotated with 10 QTLs on CHR 2, 3, 14, 15, and 16.

The annotated traits are known to be associated with body weight, milk fat yield, platelet count, total lambs born, and soft tissue depth at the GR site. The QTLs identified in each breed were categorized into various trait classes, including body weight, reproduction, milk yield, and health, and the enrichment of QTLs is shown in Fig 2. The highest percentage of QTLs annotated in the significant regions of Chanthangi were milk (37.5%), followed by reproduction (34.38%). The other classes of QTLs annotated were meat and carcass (9.38%), health (9.38%), and production (milk fat yield) (9.38%). The QTL enrichment analysis revealed the top significant (False Discovery Rate (FDR)-corrected *p*-value – 0.05) QTL total lambs born on OAR 16.

### 3.3.3. Garole sheep (gene annotation).

In this study, several putative genes were identified in the significant genomic regions located on OAR 1, 2, 3, 6, 16, and 18 (Table 3).

However, OAR 3 anchors more potential candidate genes in the GAR breed. In this study, we identified eleven putative genes associated with a wide range of characteristics, including *BBX* (teat number), *CNTNAP5* (body growth), *C2CD5* (fecundity), *KDM5A* (follicular development), *B4GALNT3* (paratuberculosis resistance), *UNC5C* (litter size), *ADCY2* and *TCERG1L* (reproduction).

**Table 1. Gene annotation in significant genomic regions in CHANTHANGI sheep.**

| CHR* | BP* | START COORDINATE | END COORDINATE | Width | Strand | Gene Name | *Pvalue* | *Qvalue* |
|------|-----|------------------|----------------|-------|--------|-----------|----------|----------|
| 2 | 190267950 | 189526649 | 190606970 | 1080322 | + | *CNTNAP5* | *4.95e-05* | *0.032* |
| 3 | 106221882 | 106222211 | 106279403 | 57193 | + | *FBLN7* | *1.29e-05* | *0.015* |
| 3 | 107958569 | 107917450 | 107989336 | 71887 | + | *TBC1D15* | *1.28e-05* | *0.015* |
| 9 | 28518494 | 28426199 | 28586681 | 160483 | + | *MTSS1* | *0.008* | *0.048* |
| 9 | 28610511 | 28596736 | 28628449 | 31714 | + | *TATDN1* | *9.16e-05* | *0.045* |
| 9 | 28651053 | 28650081 | 28651912 | 1832 | – | *TRMT12* | *0.000* | *0.048* |
| 9 | 28651053 | 28627373 | 28650325 | 22953 | – | *RNF139* | *0.000* | *0.048* |
| 14 | 35090205 | 35080140 | 35173351 | 93212 | + | *NFATC3* | *4.59e-06* | *0.009* |
| 14 | 34566935 | 34510632 | 34569231 | 58600 | – | *TPPP3* | *7.71e-05* | *0.042* |
| 14 | 34566935 | 34504860 | 34579222 | 74363 | + | *LRRC36* | *7.71e-05* | *0.042* |
| 15 | 51070453 | 51028830 | 51075998 | 47169 | – | *STARD10* | *2.34e-08* | *0.000* |
| 15 | 51107183 | 51099187 | 51353916 | 254730 | – | *FCHSD2* | *2.34e-08* | *0.000* |
| 15 | 51474825 | 51404925 | 51511918 | 106994 | + | *P2RY2* | *1.39e-08* | *0.000* |
| 15 | 51517148 | 51481919 | 51516379 | 34461 | + | *P2RY6* | *1.39e-08* | *0.000* |
| 15 | 51578641 | 51521915 | 51579398 | 57484 | + | *ARHGEF17* | *6.12e-10* | *2.57e-05* |
| 15 | 51591704 | 51585751 | 51604071 | 18321 | + | *RELT* | *3.07e-09* | *6.44e-05* |
| 15 | 51632422 | 51613275 | 51814944 | 201670 | – | *FAM168A* | *1.19e-05* | *0.015* |
| 16 | 30667706 | 30604422 | 30702581 | 98160 | + | *FGF10* | *7.11e-06* | *0.010* |
| 16 | 31588015 | 31575745 | 31651791 | 76047 | – | *NIM1K* | *4.12e-05* | *0.031* |
| 16 | 31687647 | 31660770 | 31688311 | 27542 | – | *ZNF131* | *1.85e-05* | *0.018* |
| 16 | 32196927 | 32069778 | 32366181 | 296404 | – | *GHR* | *9.62e-07* | *0.003* |

*CHR – Chromosome; *BP – Base Position.

**Table 2. Quantitative trait loci annotation in significant genomic regions in CHANTHANGI sheep.**

| CHR* | BP* | START COORDINATE | END COORDINATE | Name | Flank Marker | *q*-value |
|------|-----|------------------|----------------|------|--------------|-----------|
| 2 | 190267950 | 190457084 | 190457088 | Body weight | rs424714738 | 1.61E-03 |
| 3 | 107991182 | 108401221 | 108401225 | Body weight | rs426980328 | 0.026 |
| 14 | 35090205 | 35417680 | 35417684 | Body weight | rs417231209 | <0.05 |
| 15 | 51070453 | 50955799 | 50955803 | Milk fat yield | rs404288918 | 0.0006245 |
| 15 | 51474825 | 51832265 | 51832269 | Platelet count | rs413264572 | 5.50E-06 |
| 16 | 31588015 | 32070073 | 32070077 | Total lambs born | rs161146164 | <0.05 |
| 16 | 31654423 | 32118461 | 32118465 | Total lambs born | rs426666828 | <0.05 |
| 16 | 32569758 | 32119256 | 32119260 | Milk fat yield | rs422431427 | 0.00031338 |
| 16 | 32569758 | 32192693 | 32192697 | Body weight | rs408124092 | <0.0001 |
| 16 | 32569758 | 32192693 | 32192697 | Soft tissue depth at GR site | rs408124092 | 0.0014 |

*CHR – Chromosome; *BP – Base Position.

### 3.3.4. Quantitative trait loci (QTL) annotation and enrichment.

The present study presents the QTL annotation findings for the GAR breed, as shown in Table 4.

Our study revealed significant genomic regions annotated with 28 QTLs on CHR 1, 2, 3, 6, 18, and 22. The annotated traits are known to be associated with teat number, body weight, milk fat yield, entropion, bone area, total lambs born,

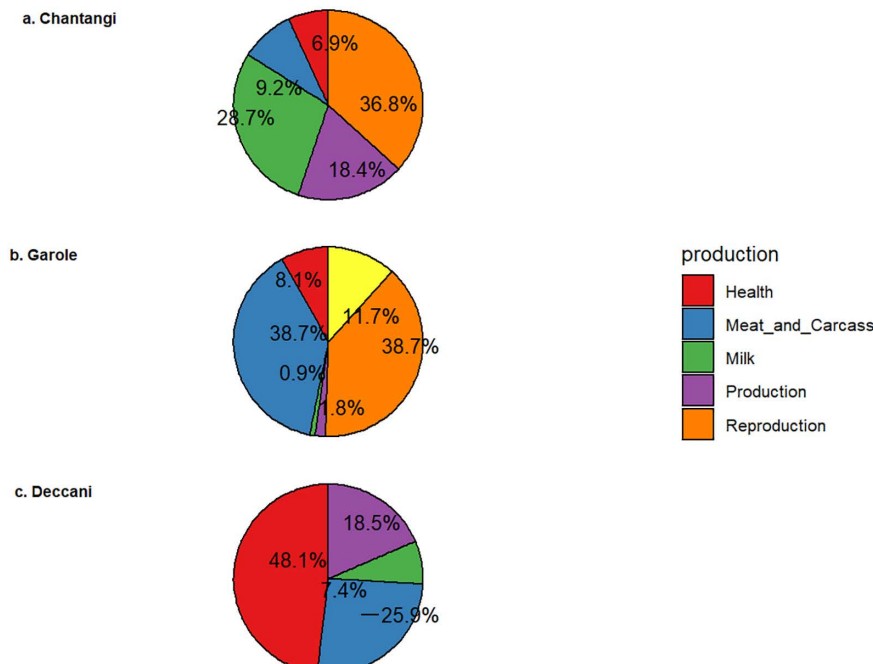

**Fig 2. Percent QTL classes annotated in three Indian sheep breeds.** This figure displays three pie charts (a, b, and **c**), each depicting the percentage contributions of various trait categories to the total identified QTLs in a specific breed. The legend positioned on the right clarifies the trait classes represented by each color: Health (red), Meat and Carcass (blue), Milk (green), Production (purple), and Reproduction (orange). **a. Chanthangi:** The QTLs in Chanthangi sheep are primarily associated with **Reproduction (36.8%),** followed by Milk (28.7%), Production (18.4%), Meat and Carcass (9.2%), and Health (6.9%). **b. Garole:** In Garole sheep, the predominant QTL categories are **Meat and Carcass (38.7%) and Reproduction (38.7%),** with lesser contributions from Health (8.1%), Production (11.7%), and Milk (1.8%). **c. Deccani:** Deccani sheep exhibit a significant prevalence of QTLs associated with **health traits (48.1%),** followed by those related to production (18.5%). Additional categories include milk (7.4%) and meat and carcass traits, which are reported as −25.9%.

**Table 3. Gene annotation in significant genomic regions in GAROLE sheep.**

| CHR* | BP* | START COORDINATE | END COORDINATE | Width | Strand | Gene | Pvalue | Qvalue |
|------|------|------|------|------|------|------|------|------|
| 1 | 173424721 | 173196985 | 173483458 | 286474 | + | BBX | 2.01e-05 | 0.015 |
| 2 | 189539157 | 189526649 | 190606970 | 1080322 | + | CNTNAP5 | 8.19e-05 | 0.042 |
| 3 | 193287569 | 193251622 | 193339539 | 87918 | + | C2CD5 | 8.81e-07 | 0.003 |
| 3 | 212967581 | 212887405 | 212973893 | 86489 | + | IQSEC3 | 6.47e-06 | 0.007 |
| 3 | 213086338 | 213068112 | 213128627 | 60516 | − | KDM5A | 1.07e-05 | 0.009 |
| 3 | 213195386 | 213175874 | 213273093 | 97220 | + | B4GALNT3 | 7.64e-06 | 0.007 |
| 6 | 29809690 | 29597358 | 30017600 | 420243 | + | UNC5C | 6.21e-06 | 0.007 |
| 16 | 65681259 | 65471620 | 65927572 | 455953 | − | ADCY2 | 7.56e-05 | 0.041 |
| 18 | 46100361 | 46097255 | 46131370 | 34116 | − | CLEC14A | 4.00e-05 | 0.024 |
| 22 | 49602008 | 49599602 | 49793506 | 193905 | − | TCERG1L | 6.19e-06 | 0.007 |

*CHR – Chromosome; *BP – Base Position.

**Table 4. Quantitative trait loci annotation in significant genomic regions in GAROLE sheep.**

| CHR* | BP* | START COORDINATE | END COORDINATE | Name | Flank Marker | q-value |
|---|---|---|---|---|---|---|
| 1 | 173424721 | 173432597 | 173432601 | Teat number | rs430157497 | <0.05 |
| 1 | 173424721 | 173534798 | 173534802 | Teat number | rs426598066 | <0.05 |
| 2 | 190019411 | 190457084 | 190457088 | Body weight | rs424714738 | 1.61E-3 |
| 3 | 192453314 | 192442586 | 192442590 | Entropion | rs411640492 | 3.19E-04 |
| 3 | 192935483 | 192442586 | 192442590 | Entropion | rs411640492 | 3.19E-04 |
| 3 | 193336800 | 193016493 | 193016497 | Entropion | rs404605277 | 1.12E-04 |
| 3 | 212967581 | 213021781 | 213021785 | Entropion | rs159913124 | 3.61E-04 |
| 3 | 213209196 | 213213768 | 213213772 | Entropion | rs418752307 | 2.81E-04 |
| 6 | 29809690 | 29344134 | 29344138 | Bone area | rs418089550 | 1.12E-5 |
| 6 | 29809690 | 29366036 | 29366040 | Milk fat yield | rs424558688 | 0.036982472 |
| 6 | 29847992 | 30096782 | 30096786 | Total lambs born | rs421635584 | <0.05 |
| 18 | 46159259 | 46086980 | 46086984 | Body weight | rs426187704 | <0.05 |
| 18 | 46100361 | 46086980 | 46086984 | Body depth | rs426187704 | <0.01 |
| 18 | 46062053 | 46086980 | 46086984 | Body weight | rs426187704 | <0.05 |
| 18 | 46062053 | 46086980 | 46086984 | Body length | rs426187704 | <0.05 |
| 18 | 46159259 | 46086980 | 46086984 | Body circumference | rs426187704 | <0.05 |
| 18 | 46159259 | 46086980 | 46086984 | Body depth | rs426187704 | <0.01 |
| 18 | 46159259 | 46086980 | 46086984 | Rump width | rs426187704 | <0.05 |
| 18 | 46159259 | 46087004 | 46087008 | Body weight | rs404696179 | <0.05 |
| 18 | 46159259 | 46087004 | 46087008 | Body length | rs404696179 | <0.01 |
| 18 | 46100361 | 46087004 | 46087008 | Body height | rs404696179 | <0.01 |
| 18 | 46159259 | 46087004 | 46087008 | Body circumference | rs404696179 | <0.05 |
| 18 | 46062053 | 46087004 | 46087008 | Body height | rs404696179 | <0.05 |
| 18 | 46159259 | 46087646 | 46087650 | Chest width | rs405457403 | <0.05 |
| 18 | 46159259 | 46087646 | 46087650 | Shin circumference | rs405457403 | <0.05 |
| 18 | 46159259 | 46087646 | 46087650 | Body depth | rs405457403 | <0.05 |
| 18 | 47232764 | 47716852 | 47716856 | Haemonchus contortus resistance | rs428856771 | 3.23E-4 |
| 22 | 49796078 | 49873766 | 49873770 | Lambs born alive | rs402605786 | 9.79E-3 |

*CHR – Chromosome; *BP – Base Position.

body depth, body length, body circumference, shin circumference, body weight, body height, chest width, rump width, lambs born alive and hemonchus contortus resistance. Furthermore, the annotated QTLs are categorised into different classes, and the enrichment of QTLs is depicted in Fig 2A. Garole's most significant genomic regions had the greatest proportion of QTLs associated with health (38.74%) and production (38.74%). The remaining percentage of QTL classes annotated in the breed were reproduction (11.71%), exterior (8.11%), milk (1.8%) and meat and carcass (0.9%). An analysis of QTL enrichment revealed the three most significant QTLs (False Discovery Rate (FDR)-corrected $p$-value < 0.05) associated with total lambs born, teat number, and entropion on CHR 6, 1, and 3.

**3.3.5. Deccani sheep (gene annotation).** The gene annotation findings are summarised in Table 5.

The gene annotation of the significant genomic regions revealed fourteen potential candidate genes related to a variety of traits in this sheep, including *BUB1* (oocyte and follicle development), *POU2F1* (wool colour), *CHRM2* (hair follicle growth), *DOCK3* (muscle development), and *MAPKAPK3* (heat stress response). These genomic regions harbour candidate genes associated with various traits and are located on CHR 1, 2, 3, 4, and 19, with the largest number of significant genomic regions on CHR 19.

**Table 5. Gene annotation in significant genomic regions in DECCANI sheep.**

| CHR* | BP* | START COORDINATE | END COORDINATE | width | strand | Gene | p-value | q-value |
|---|---|---|---|---|---|---|---|---|
| 1 | 119053600 | 119027142 | 119081828 | 54687 | + | MAEL | 9.08e-09 | 4.24e-05 |
| 1 | 131092601 | 130838877 | 131151750 | 312874 | + | APP | 3.55e-07 | 0.000 |
| 1 | 119491972 | 119428361 | 119624526 | 196166 | + | POU2F1 | 1.33e-05 | 0.007 |
| 3 | 104706333 | 104671436 | 104716760 | 45325 | – | MALL | 9.21e-13 | 1.29e-08 |
| 3 | 104775041 | 104716342 | 104784266 | 67925 | – | NPHP1 | 9.21e-13 | 1.29e-08 |
| 3 | 104885280 | 104870169 | 104915494 | 45326 | – | BUB1 | 1.00e-13 | 4.20e-09 |
| 3 | 212967581 | 212887405 | 212973893 | 86489 | + | IQSEC3 | 0.000 | 0.044 |
| 3 | 104917562 | 104914948 | 105286674 | 371727 | + | ACOXL | 1.64e-11 | 1.15e-07 |
| 4 | 101967194 | 101379345 | 102196659 | 817315 | + | CHRM2 | 1.64e-11 | 1.15e-07 |
| 19 | 49657211 | 49447036 | 49738575 | 291540 | – | DOCK3 | 6.78e-06 | 0.005 |
| 19 | 49778830 | 49759229 | 49789524 | 30296 | – | MAPKAPK3 | 5.45e-06 | 0.004 |
| 2 | 52731947 | 52731991 | 52734550 | 2560 | + | HINT2 | 5.26e-07 | 0.001 |
| 2 | 52790093 | 52790313 | 52801134 | 10822 | – | RGP1 | 5.26e-07 0.001 | 0.001 |
| 2 | 52790093 | 52790904 | 52793100 | 2197 | + | MSMP | 5.26e-07 0.001 | 0.001 |

*CHR – Chromosome; *BP – Base Position.

**3.3.6. Quantitative trait loci (QTL) annotation and enrichment.** The findings outlined in Table 6 detail the results of QTL annotation in Deccani sheep.

The study has identified significant genomic regions annotated with 13 QTLs on CHR 1, 2, 3, 4, and 21. The QTLs are associated with the following traits: somatic cell count, Entropion, Monocyte number, Footrot susceptibility, Pneumonia susceptibility, Body length, Body weight, water-holding capacity, and horn type. Furthermore, the annotated QTLs are grouped into various categories, and their distribution is illustrated in Fig 2. The regions with the most significant impact on the evolution of the Deccani breed had the highest percentage of QTLs annotated for exterior traits (48.15%), followed by health traits (25.93%). The other classes of QTLs annotated were production (18.52%), and meat and carcass (7.41%).

**Table 6. Shows the quantitative trait loci annotation results in significant genomic regions in DECCANI sheep.**

| CHR* | BP* | start_pos | end_pos | Name | FlankMarker | q-value |
|---|---|---|---|---|---|---|
| 1 | 131092601 | 130616460 | 130616464 | somatic cell count | rs409795528 | <0.05 |
| 3 | 212967581 | 213021781 | 213021785 | Entropion | rs159913124 | 3.61E-04 |
| 3 | 212967581 | 213213768 | 213213772 | Entropion | rs418752307 | 2.81E-04 |
| 4 | 101967194 | 101960285 | 101960289 | Monocyte number | rs399619443 | 2.55E-6 |
| 4 | 101967194 | 102026928 | 102026932 | Footrot susceptibility | rs416121047 | 2.84E-6 |
| 4 | 101967194 | 102039511 | 102039515 | Pneumonia susceptibility | rs403394816 | <0.10 |
| 21 | 41665172 | 41243033 | 41243037 | Body length | rs406947061 | <0.05 |
| 2 | 52731947 | 52720586 | 52720590 | Body weight | rs426272889 | <0.05 |
| 2 | 52790093 | 52737532 | 52737536 | Body weight | rs160159557 | <0.05 |
| 2 | 52790093 | 52848103 | 52848107 | Water holding capacity | rs420647640 | <0.05 |
| 2 | 116594826 | 116133033 | 116133037 | Horn type | rs416045857 | <0.05 |
| 2 | 116594826 | 116839657 | 116839661 | Horn type | rs405044765 | <0.05 |
| 2 | 116953479 | 116872298 | 116872302 | Horn type | rs402400571 | <0.05 |

*CHR – Chromosome; *BP – Base Position.

The QTL enrichment analysis identified the top-most significant QTLs (False Discovery Rate (FDR)-corrected *p*-value <0.05) on CHR 1,2, and 3, controlling traits such as somatic cell count, horn type, Entropion, Body weight, and Body length (Fig 3). Furthermore, the signatures of selection in the genomes of the studied breeds are shown in Fig 4.

In addition, the summary of putative genes is presented in Table 7.

## 4. Discussion

This study used an SNP-based, decorrelated composite of multiple signals (DCMS) to identify the genetic basis of important economic and adaptation traits that influence the genomic landscape in three indigenous Indian sheep breeds: Chanthangi, Garole, and Deccani. It uncovered novel, breed-specific genomic regions, genes, and quantitative trait loci (QTLs) under selection in these breeds. The identified regions provided key insights into the molecular mechanisms responsible for traits such as growth, reproduction, wool production, disease resistance, and environmental adaptation. This research offers a strong foundation for developing genetic improvement strategies and conservation programs for these valuable indigenous breeds.

### 4.1. Principal component analysis (PCA)

Principal Component Analysis (PCA) demonstrated distinct genetic differentiation and clustering patterns among the studied Indian sheep breeds (Fig 1). This apparent population stratification, however, was inconsistent with previous findings

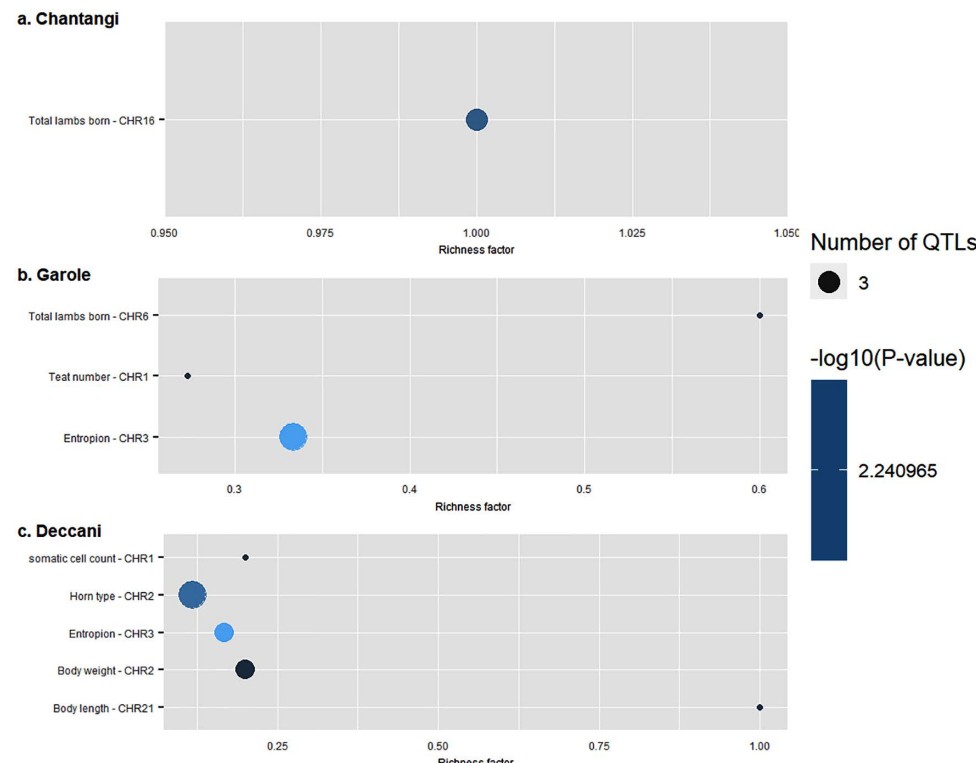

**Fig 3. QTL classes enriched in three Indian sheep breeds.** This figure shows three scatter plots (a, b, and c) illustrating enriched QTL classes in genomic regions selected for a specific breed. The y-axis lists enriched trait categories and their corresponding chromosomes. The x-axis displays the 'Richness factor,' which indicates the level of enrichment for each QTL class. Each bubble on the plot represents an enriched QTL class. The size of each bubble corresponds to the 'Number of QTLs' identified for that trait, and the color reflects the statistical significance of the enrichment, measured by -log10(*p*-value). **a. Chanthangi**: Shows enrichment for 'Total lambs born' on Chromosome 16 (CHR16). **b. Garole**: Shows enrichment for 'Total lambs born' on CHR6, 'Teat number' on CHR1, and 'Entropion' on CHR3. **c. Deccani**: Indicates enrichment for 'Somatic cell count' on CHR1, 'Horn type' on CHR2, 'Entropion' on CHR3, 'Body weight' on CHR2, and 'Body length' on CHR21.

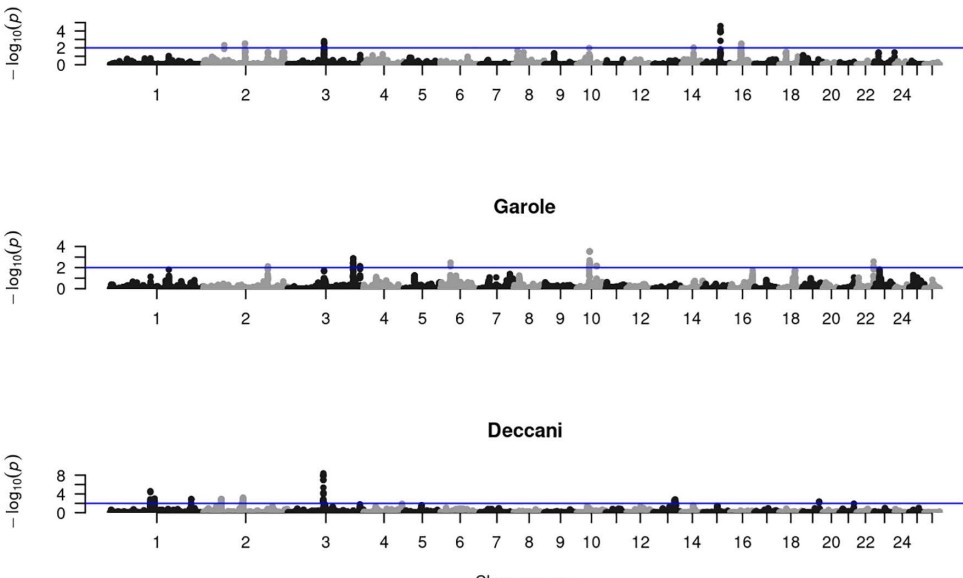

**Fig 4. Manhattan plot showing signatures of selection in three Indian sheep breeds.** This figure includes three separate Manhattan plots, each depicting a sheep breed: a) Chanthangi, b) Garole, and c) Deccani. Each plot shows the genomic regions under selection pressure. The x-axis of all plots denotes the chromosomal position, with individual chromosomes (1 to 26) arranged sequentially and distinguished by alternating grey and black hues. Whereas, the y-axis of the Chanthangi and Garole plots ranges from 0 to 4, and in the Deccani plot, the y-axis scale extends from 0 to 8, which shows the negative base-10 logarithm of the *p*-value (-log10(*p*-value)), derived from the DCMS statistic. The horizontal blue line in each plot indicates the genome-wide significance threshold; SNPs clustering above this threshold highlights genomic regions that have undergone significant selection sweeps. Chanthangi shows prominent selection signals on chromosome 15, Garole on chromosomes 3 and 10, and Deccani on chromosome 3.

**Table 7. Summary of putative genes identified in Indian sheep.**

| CHR* | BP* | START COORDINATE | END COORDINATE | Gene | Trait |
|---|---|---|---|---|---|
| 2 | 189539157 | 189526649 | 190606970 | *CNTNAP5* | Growth and muscle development |
| 3 | 193287569 | 193251622 | 193339539 | *C2CD5* | Reproduction |
| 9 | 28518494 | 28426199 | 28586681 | *MTSS1* | Disease resistance |
| 3 | 213195386 | 213175874 | 213273093 | *B4GALNT3* | Disease resistance |
| 9 | 28651053 | 28650081 | 28651912 | *TRMT12* | Adaptation |
| 3 | 104885280 | 104870169 | 104915494 | *BUB1* | Reproduction |
| 6 | 29809690 | 29597358 | 30017600 | *UNC5C* | Reproduction |
| 14 | 34566935 | 34504860 | 34579222 | *LRRC36* | Domestication |
| 19 | 49657211 | 49447036 | 49738575 | *DOCK3* | Growth and muscle development |
| 19 | 49778830 | 49759229 | 49789524 | *MAPKAPK3* | Adaptation |
| 15 | 51517148 | 51481919 | 51516379 | *P2RY6* | Wool trait |
| 16 | 30667706 | 30604422 | 30702581 | *FGF10* | Wool trait |

by Ahmad et al. (2021) [40]. who reported a lack of distinct clustering among the breeds they investigated based on the first principal component. This divergence in results may be attributed to Ahmad et al.'s inclusion of a broader range of breeds, which likely introduced additional genetic variation into their analysis.

## 4.2. Chanthangi sheep

### 4.2.1. Gene annotation.

India's diverse sheep production systems are adapted to a wide spectrum of climates, ranging from the cold Northern Himalayan foothills to the hot and arid Central Peninsular region and the humid Eastern region. Reflecting these varied environmental pressures, our findings indicated that none of the identified candidate genes within the detected genomic regions showed significant selection signatures consistently across all three analyzed breeds. Nonetheless, the identified genomic regions harbored potential candidate genes associated with a broad spectrum of traits, including body growth, reproduction, fecundity, heat-stress response, host resistance to bacterial infections, domestication, environmental adaptation, hair follicle growth, meat quality, and litter size. The minimal overlap observed in significant genomic regions among the breeds strongly suggests the prevalence of breed-specific selection processes driven by local adaptations throughout their evolutionary history.

The present study identified *CNTNAP5* (Contactin-associated protein-like 5B) as a significant genomic region located on CHR 2, and as a protein-coding gene. This gene belongs to the neurexin family and plays an important role as a cell adhesion molecule and receptor in the nervous systems of most vertebrates. The mechanism of action of *CNTNAP5* is similar to that of the brain-derived neurotrophic factor (*BDNF*) gene, which influences body mass index. Like other epidermal growth factors, it potentially influences growth by enhancing the functions of the brain that regulate growth [41,42]. Genomic investigations of morphological traits in Sudanese goats revealed that the *CNTNAP5* gene, located on chromosome 2, was associated with the bicostal diameter [43].

The indigenous sheep of India are known for resistance to infectious agents. Our findings also identified specific genomic regions in Chanthangi sheep that harbor the *MTSS1* gene on chromosome CHR9, which is associated with resistance to bacterial infections. These findings were consistent with the breed characteristics of Chanthangi. The bacteria's endotoxins disrupt the endothelial lining in the host cell and cause infection. However, the *MTSS1* gene alters the host endothelium and improves its integrity, thereby preventing infections [44]. The DCMS analysis also revealed significant genomic regions containing the *STARD10* gene, which is reportedly involved in resistance to paratuberculosis [45]. It regulates bile acid metabolism by modulating the PPARα-mediated pathway [46]. The study identified an essential candidate gene, *LRRC36*, associated with domestication in sheep. *LRRC36* is a leucine rich repeat containing 36 gene differentially expressed in testis and lung tissues. Studies on the structural variant landscape in sheep and goats revealed that the *LRRC36* gene was involved in the early domestication [47]. Another candidate gene, *TRMT12*, a tRNA methyltransferase, is fundamental for tRNA modification, which is vital for regulating mRNA translation and protein homeostasis. Although direct studies on its role in environmental adaptation in sheep are currently absent, its molecular function suggests a plausible contribution to cellular resilience against diverse stressors [48,49]. By ensuring efficient protein synthesis, *TRMT12* could mediate cellular adaptation under challenging environmental conditions, making it a compelling candidate gene in indigenous sheep breeds.

Chanthangi sheep are primarily reared by the Changpa, a nomadic tribe of the Northern hills, along with Chanthangi goats. This dual-purpose sheep breed, maintained for both wool and mutton, contributes significantly to the livelihoods of nomadic tribes. Shepherd's income mainly derives from selling wool and meat. Our study highlighted critical candidate genes associated with hair follicle development, including *P2RY6* (OAR 15) and *FGF10* (OAR 16). A gene expression study in mice demonstrated the role of *P2RY6* in controlling the *MEK1/2*–b-catenin signalling pathways and the *MST–LATS1–YAP* signal cascades, indicating its function in keratinocyte growth and development [50]. Moreover, a gene expression study on coarse sheep fetal skin showed that *FGF10* was involved in hair follicle development via competitive adsorption of miR-184, which produces *FGF10* from chi-circRNA-0001141 [51].

We also identified genes under selection in Chanthangi sheep, including *ARHGEF17* and *NIM1K,* which are known to be related to meat quality. Weighted single-step genome-wide association studies (GWAS) revealed a significant association between the *ARHGEF17* gene and meat traits in the Chinese yellow-feathered chicken population [52]. *ARHGEF17* is a key component of actin cytoskeletal organization and enhances guanine nucleotide exchange factor

activity. Transcriptomic and epigenomic studies in the pectoral muscles of chicken embryos demonstrated *NIM1K*'s significant role in muscle fibre formation. *NIM1K* enables ATP binding, magnesium ion binding, and protein serine/threonine kinase activity and is involved in protein phosphorylation [53]. Notably, several of these genes, such as *CNTNAP5, DOCK3, TCERG1L, BUB1, UNC5C, C2CD5, BBX, TPPP3, FGF10, POU2F1, FAM168A, MTSS1, B4GALNT3, TRMT12, MAPKAPK3,* and *LRRC36,* were not detected in earlier window-based DCMS analyses [20]**.** This highlights the advantage of SNP-level resolution for mapping breed-specific loci that are potentially overlooked by broader window-based approaches.

**4.2.2. Quantitative trait loci (QTL) annotation and enrichment.** The DCMS analysis revealed important QTL regions associated with body weight on OAR 2, 3, 14, and 16. The flanking markers or SNPs in the QTL regions, include rs424714738, rs426980328, rs417231209, and rs408124092, respectively. A genome-wide association study in Australian Merino sheep identified a significant QTL region on OAR2 associated with body weight. The study also found an orthologous region on OAR6, similar to human and bovine genomic regions associated with height and weight [54]. Our study also identified the same QTL as significant across the studied breeds. Zhang et al., 2016 identified *MEF2B* and *TRHDE* gene polymorphisms linked to body weight at 4 months of age in New Ujumqin Sheep on OAR 3 [55]. DCMS analysis revealed the same QTL region under selection in Indian populations. In Baluchi sheep, GWAS confirmed the QTL peak associated with 8-month body weight [56]. We identified a QTL on OAR 14 in Chanthangi sheep. Armstrong et al., (2018) investigated genetic polymorphisms in the *GHR* gene and demonstrated associations with carcass traits in grazing Texel sheep; the same QTL peak was observed in the Chanthangi sheep [57].

We also identified QTL related to platelet count and milk yield on OAR15 in Chanthangi sheep. Gonzalez et al., (2013) reported QTL association with mean corpuscular haemoglobin concentration (MCHC, $P = 6.2 \times 10 - 14$) and decreased mean corpuscular volume in Columbia, Polypay, and Rambouillet sheep [58]. DCMS analyses also identified platelet count QTL in significant genomic regions of Chanthangi sheep. Genomic investigation in the crossbred dairy sheep population revealed milk yield QTL regions [59]. We found the same QTL regions in Chanthangi.

## 4.3. Garole sheep

The Garole sheep were renowned for their petite stature, remarkable fertility, superior meat quality, and ability to thrive in the saline marshlands of the hot, humid Sundarban region in West Bengal. This breed was also found across southern Bangladesh. Its small size resulted from adaptation to a harsh climate. Although genetic potential was limited, these sheep could be raised with minimal input management (Pan & Sahoo, 2008) [60]. The breed was highly valued for its fecundity-related genes.

**4.3.1. Gene annotation.** The gene annotation revealed candidate genes associated with essential traits in Garole sheep. We identified the *BBX* gene on OAR1, which is associated with supernumerary teats in sheep. Interestingly, a genome-wide association study of Wadi sheep in China identified candidate genes linked to the supernumerary nipple phenotype, including BBX [61]. *BBX* participates in cell proliferation and breast tumors, producing a high-mobility group domain (*HMG*) transcription factor observed in breast tumors [62]. DCMS analysis revealed the *CNTNAP5* gene on OAR 2, which regulates body growth, and it is also annotated in the Chanthangi [43]. The same genomic region was selected for in both breeds. We identified candidate genes on OAR 3 in GAR sheep, including *C2CD5, IQSEC3, KDM5A,* and *B4GALNT3*. Expression studies in sheep oviductal mRNAs and lncRNAs highlighted C2CD5 [63], which is involved in calcium-dependent phospholipid binding and insulin receptor signalling, linked to prolificacy and litter size, aligning with the breed history. *KDM5A* plays a role in zygote genome activation in cloned goat embryos [64]. DCMS analysis also highlighted *KDM5A* in GAR sheep. *UNC5C,* another candidate gene under selection, was confirmed to influence litter size in Hu sheep [65]. *UNC5C* is a part of the *UNC-5* family of netrin receptors and is associated with conception rate in cattle [66]. Multiple genome-wide studies confirmed its role in sheep litter size [19,67,68].

The *ADCY2* gene, known for pleiotropic effects, produces *cAMP*, a signalling molecule in response to G protein signal transduction, leading to increased *IL-6* and influencing Brucellosis [69,70]. It also enhances estradiol expression, which is important for reproduction [71]. We identified *ADCY2* in our study.

**4.3.2. Quantitative trait loci (QTL) annotation and enrichment.** DCMS analysis identified QTL regions associated with teat number, body weight, body measurements, entropion, resistance to the Hemonchus infection, total lambs born, and total lambs alive.

We identified the peak QTL region at 201.6 cM; including markers rs430157497 and rs426598066. This region on OAR1, surrounding *BBX* and *CD47*, was associated with supernumerary teats in Wadi sheep in China [61]. A significant region on OAR6 was linked to bone area in Garole sheep; prior GWAS in Scottish Blackface lambs identified genes influencing bone density and area [72].

Garole sheep are known for their prolificacy and are valued for their litter size. Studies have confirmed genomic regions on OAR 6 and 22 that are potentially related to litter size in multiple breeds, including Wadi, Hu, Icelandic, Finnsheep, Romanov, and Lori-Bakhtiari [73,74]. Genes in these regions, including *BMPR1B*, *CTNNB1*, and *LHCGR*, significantly affect litter size. Remarkably, this QTL aligns with Garole's breeding history.

A study on the Somatostatin Receptor Subtype 1 (SSTR1) gene polymorphism confirmed an association with growth traits in Hulun Buir Sheep. We observed QTL on OAR18 related to various body measurements such as body weight, length, circumference, rump width, body depth, chest width, and shin circumference [75].

## 4.4. Deccani sheep

**4.4.1. Gene annotation.** The Deccani breed is the only pure-black, coarse-wool breed indigenous to India. Our study identified significant genomic regions harboring candidate genes *MAEL, APP*, and *POU2F1*. *POU2F1* regulated melanin production by binding with *SLC7A11* and inhibiting its function [76]. It also affected other genes (*SLC7A11, MITF, SLC24A5, MC1R,* and *ASIP*), which impact melanin production and wool colour [77], supporting Deccani's unique traits. Another region on OAR3 contained *MALL, NPHP1, BUB1, IQSEC3,* and *ACOXL*. *MALL* and *NPHP1* regulated cell division and cell-matrix adhesion signalling. The *BUB1* gene encodes a protein involved in the DNA damage response. *IQSEC3* controlled the small GTPase-mediated signal transduction, and *ACOXL* was associated with fatty acid beta-oxidation via acyl-CoA oxidase. Previous genomic studies also confirmed the role of this region in immune response [78]. Interestingly, we also identified this region.

We detected a region on OAR2 containing *HINT2, RGP1*, and *MSMP*, genes controlling growth and fat deposition. GWAS in Ethiopian sheep linked these genes to fat deposition and growth [79]. *HINT2* is involved in growth and fat deposition by facilitating the hydrolysis of various bonds, such as C-O, C-N, and C-C. *RGP1* gene promoted vesicle transport to the trans-Golgi network, and *MSMP* encoded a member of the beta-microseminoprotein family. Other studies have reported the associations with fat deposition and growth [80,81].

On OAR 14, *DOCK3* was a candidate gene associated with muscle development in cattle. Studies in Bashbay sheep highlighted the role of *DOCK3* in skeletal muscle development [82]. The *DOCK* gene family comprises guanine nucleotide exchange factors that play a crucial role in myoblast fusion and migration, expressed in skeletal muscle through *RAC1* and *WAVE* signalling pathways [83].

**4.4.2. Quantitative trait loci (QTL) annotation and enrichment.** The QTL analysis uncovered regions under selection controlling somatic cell count, entropion, footrot, pneumonia susceptibility, monocyte number, body weight, body length, water-holding capacity, and horn length. DCMS analysis identified a region associated with somatic cell count. A GWAS reported significant associations with milk production, composition, coagulation, and cheese traits, including somatic cell count, in Assaf and Churra dairy sheep [84].

Monocytes are a type of white blood cell vital to the immune response. Differentiated monocytes play a role in phagocytosis and antigen presentation, thereby supporting adaptive immunity. In sheep, monocytes and macrophages counteract

the septicemic conditions caused by Coxiella burnetii and lentivirus [85]. GWAS in Rambouillet, Polypay and Columbia sheep identified loci associated with the monocyte count on multiple chromosomes, including OAR 1, 2, 3, 4, 9, 10, 15, and 16. We found the loci on OAR4 associated with the monocyte count in the Deccani sheep genome.

Two loci on chromosome 4 associated with foot rot and pneumonia susceptibility have been identified. A GWAS study in Katahdin, Blackbelly, and European-influenced crossbred sheep identified loci on chromosome 4 associated with foot rot susceptibility [86].

The genomic analyses revealed a significant association between body length and body weight on chromosomes 21 and 2. Notably, previous genome-wide association studies (GWAS) in Qira Black and German Merino sheep have also demonstrated associations between QTL regions and body measurements, including body weight traits [87].

The present study identified two robust genomic associations on chromosome 2 in Deccani sheep: one related to water-holding capacity (a meat quality trait) and the other to horn type. Deccani sheep are valued by consumers for their superior meat quality and distinctive physical characteristics, especially horn morphology, which is unique to the breed. Consistent with our findings, a GWAS on Colombian Creole hair sheep also reported strong associations between genomic regions on chromosome 2 and water-holding capacity in meat [88]. Furthermore, another genomic study elucidated the genomic basis of horn polymorphism in wild Soay sheep, revealing a strong association with regions on chromosome 2 [89].

The findings align with previous genomic research on Chanthangi sheep [18] reinforcing the identification of significant genomic regions and their associations. Similar studies [17,18,20], highlight candidate genes influencing growth, wool production, and disease resistance. These results validate existing genomic insights and contribute to understanding the genetic foundations of key traits. Consistency across studies emphasizes these region's importance in sheep breeding programs.

This study uncovered breed- and trait-specific signature genomic regions and candidate genes in Indian sheep. Our DCMS analysis showed minimal overlap with regions reported by Muthuswamy et al. (2025). Overlapping regions included *STARD10, FCHSD2, P2RY2*, and *P2RY6* on *OAR* 15 in Changthangi, *UNC5C* on *OAR* 6 in Garole, and *APP* on *OAR* 1 in Deccani sheep. Concordance across both SNP-centric and window-based DCMS approaches underscores the robustness of the selection signals at these loci. However, the limited overlap also suggests that each analytical method captures unique aspects of the genomic landscape under selection: window-based methods may be more sensitive to broader regions with weaker, cumulative signals, and SNP-centric approaches offer greater precision, pinpointing individual loci with stronger effects.

The SNP-based de-correlated composite of multiple signals (DCMS) approach used in this study revealed high-resolution breed-specific selection signatures in Indian sheep. This method provided essential insights into the mechanisms underlying adaptation to challenging environments and the development of significant economic traits. The findings have important implications for the sustainable management and genetic improvement of indigenous livestock. The candidate genes and Quantitative Trait Loci (QTLs) identified in this study serve as key targets for marker-assisted selection to develop more productive and resilient animals. This study also emphasizes the importance of the native genetic resources, supporting conservation efforts to maintain their adaptive traits and diversity for future generations.

The results of this research align with prior investigations in identifying candidate genes and genomic regions linked to body weight, litter size, and disease resistance [17,18,20]. The PCA analysis in this study showed distinct clustering, contradicting the previous study on Indian sheep [18]. Furthermore, the minimal overlap between our identified selection signatures and those reported in earlier window-based DCMS analyses [20] for these breeds underscores a crucial distinction: our SNP-centric approach provides superior resolution, facilitating the identification of loci that may have been missed by broader genomic interval analyses, rather than indicating a disagreement in the underlying biological phenomena.

A key limitation of the present study is the absence of direct experimental validation for the identified candidate genes and quantitative trait loci. While our computational genomic approach provides strong evidence for selection signatures and putative functional associations, the precise biological mechanisms and phenotypic effects of these loci were not experimentally confirmed. Future research involving functional genomics experiments, such as gene editing or targeted expression studies, and detailed phenotypic validation on larger populations, would be crucial to conclusively establish the roles of these candidate regions in Indian sheep breeds.

## 5. Conclusions

We identified 51 putative QTL regions encompassing 45 protein-coding genes in Indian sheep breeds under selection and adapted to unique and distinct local climates. The putative QTLs and functional candidate genes revealed in the study reflect the historical adaptation and selection pressures experienced by these breeds. The identified signature genomic regions were associated with economic traits, including body growth, fecundity, heat stress response, domestication, environmental adaptation, hair follicle growth, horn type, and meat quality yield. Importantly, our analyses uncovered breed- and trait-specific genomic regions, such as those linked to hair follicle growth in Chanthangi, fecundity and lamb survivability in Garole, and meat quality traits in Deccani, which had not been previously reported in window-based studies. These results provide new insights into the molecular mechanisms underlying important traits and local adaptation in Indian sheep. Additionally, our comprehensive approach provides valuable genomic resources for future breeding programs to improve the genetics of indigenous Indian sheep breeds and conserve them. While these findings offer valuable insights and expand genomic resources, further experimental validation is warranted to confirm the functional roles of the identified candidate genes and QTLs. In conclusion, our SNP-based DCMS analysis revealed novel genes and QTLs associated with economically important traits in Indian sheep breeds, expanding the genomic resources available for their improvement and conservation. This work lays a foundation for precision breeding and sustainable management of indigenous livestock adapted to challenging environments.

## Acknowledgments

The first and second authors duly acknowledge the Sri Venkateswara Veterinary University, Tirupati, Andhra Pradesh, for its support in conducting this study.

## Author contributions

**Conceptualization:** Sapna Nath, Satish Kumar Illa.

**Data curation:** Sapna Nath, Satish Kumar Illa.

**Formal analysis:** Destaw Worku, Sapna Nath, Satish Kumar Illa, Sabyasachi Mukherjee, Anupama Mukherjee, Vinod Kumar Yata.

**Methodology:** Destaw Worku, Sapna Nath, Satish Kumar Illa.

**Supervision:** Destaw Worku.

**Visualization:** Destaw Worku.

**Writing – original draft:** Sapna Nath, Satish Kumar Illa.

**Writing – review & editing:** Destaw Worku, Sabyasachi Mukherjee, Anupama Mukherjee, Vinod Kumar Yata.

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
