## [Decision Letter · Decision Letter 0]

10 Feb 2025

Dear Dr. Worku,

Thank you for submitting your manuscript to PLOS ONE. After careful consideration, we feel that it has merit but does not fully meet PLOS ONE’s publication criteria as it currently stands. Therefore, we invite you to submit a revised version of the manuscript that addresses the points raised during the review process.

Dear Dr., Destaw Worku Worku

Thank you for submitting your manuscript to PLOS ONE. After careful consideration, we have decided that your manuscript needs Major Revision.

**Comment editor**

Additional Editor Comments:

It is necessary to read the reviewers’ comments and make all the required notes carefully.

Kind regards,

Prof. Lamiaa Mostafa Radwan, Ph.D.

Academic Editor

PLOS ONE

**Reviewer 1**

The study addresses an important topic in animal genetics, focusing on Indian sheep breeds, which are of economic and cultural importance. The identification of candidate genes and QTLs has significant implications for improving breed performance and conservation strategies.

However, the manuscript requires some clarifications and revisions to enhance its scientific rigor and readability.

The manuscript is well-organized overall, but some sections (e.g., methods and results) require more detailed explanations.

Certain paragraphs in the introduction and discussion are redundant and could be condensed.

Specific Comments:

The abstract provides a good summary but lacks specific details about key findings. Including numerical results or specific genes/QTLs would enhance its informativeness.

The introduction is comprehensive but overly descriptive in some parts. Focus more on the knowledge gap and the study's unique contributions.

Include more references to recent studies on selection signals and genomic studies in sheep from diverse geographical contexts.

Describe the data quality control steps in more detail. For example, how were outlier SNPs or missing data handled?

Clarify the composite selection signal (CSS) methodology. Provide justification for using this approach over others (e.g., iHS, XP-EHH).

Were any statistical corrections applied to account for population structure or relatedness among individuals?

The presentation of the results is clear, but:

Include figures or tables summarizing key candidate genes or QTLs identified.

Highlight specific loci and their putative roles in traits relevant to Indian sheep (e.g., drought tolerance, disease resistance).

Consider including a heatmap or Manhattan plot for better visualization of selection signals.

The discussion lacks depth in interpreting biological functions of identified genes. Provide insights into how these genes/QTLs align with known traits in sheep.

Compare findings with results from other studies (e.g., on sheep from different regions or breeds).

Discuss potential limitations of the study, such as sample size or marker density, and how they might affect the robustness of the conclusions.

Figures are clear but lack detailed legends. Add more context to explain what the reader should infer from each figure.

Tables summarizing selection signals should include effect sizes and p-values where relevant.

**Reviewer 2**

Authors are requested to revise the manuscript as per the comments or suggestions put into the manuscript file uploaded in the system, also expected to address the issues related to Discussion section as under:

1. Link Between Genomic Regions and Local Adaptation: While the discussion mentions the environmental adaptations of the breeds, it could further elaborate on how these adaptations might have influenced the specific genomic regions under selection. For example, more direct comparisons to how the genetic variations might correlate with the climate or geographical features of the areas where these breeds are found could enhance the reader's understanding of the adaptive significance of these genes.

2. Consistency in Gene References: Some genes are mentioned repeatedly (e.g., CNTNAP5), but the context of how they were identified in different breeds could be more explicitly connected to avoid confusion. For instance, you might clarify how CNTNAP5's role in body growth is shared across breeds like CHA and GAR, yet each breed’s unique genomic context could influence the gene's effect.

3. Discussion on Statistical Methodology: The use of the DCMS method is discussed, but more detailed insight into why this method improves the resolution power of the genomic signals could help readers understand the methodological benefits. Briefly touching on the weaknesses of other statistical approaches and how the DCMS method overcomes them would strengthen the argument.

4. Gene Functionality and Mechanisms: While candidate genes are identified, more information on their molecular mechanisms or the processes through which they impact the phenotypic traits (e.g., how MTSS1 influences bacterial resistance) could help provide a clearer understanding of their biological relevance.

5. Comparison with Other Studies: The discussion of prior research, such as the studies from Ahmad et al. (2021) and Zhang et al. (2016), provides useful context. However, it would be helpful to briefly explain how the findings in the current study either align or contrast with these studies, reinforcing the study’s contribution to the field.

**Reviewer 3**

1. The manuscript, especially the discussion part is exceedingly long with redundancy in a few parts which is repeated in the results and discussion section.

2. Check the spelling “Chantangi” which is wrongly written.

3. Concise the discussion part and avoid redundancy.

We look forward to receiving your revised manuscript.

Kind regards,

Lamiaa Mostafa Radwan, Ph.D.

Academic Editor

PLOS ONE

**Journal Requirements:**

2. In the online submission form, you indicated that The genotypic data and other material related to this study are available with the corresponding author.

3. Ethics statement does not appear in the manuscript file:

Please include your full ethics statement in the ‘Methods’ section of your manuscript file. In your statement, please include the full name of the IRB or ethics committee who approved or waived your study, as well as whether or not you obtained informed written or verbal consent. If consent was waived for your study, please include this information in your statement as well.

4. Please include a copy of Table 3 which you refer in your text.

**Additional Editor Comments:**

Dear Dr., Destaw Worku Worku

Thank you for submitting your manuscript to PLOS ONE. After careful consideration, we have decided that your manuscript needs Major Revision.

Comment editor

Additional Editor Comments:

It is necessary to read the reviewers’ comments and make all the required notes carefully.

Kind regards,

Prof. Lamiaa Mostafa Radwan, Ph.D.

Academic Editor

PLOS ONE

**Reviewer 1**

The study addresses an important topic in animal genetics, focusing on Indian sheep breeds, which are of economic and cultural importance. The identification of candidate genes and QTLs has significant implications for improving breed performance and conservation strategies.

However, the manuscript requires some clarifications and revisions to enhance its scientific rigor and readability.

The manuscript is well-organized overall, but some sections (e.g., methods and results) require more detailed explanations.

Certain paragraphs in the introduction and discussion are redundant and could be condensed.

Specific Comments:

The abstract provides a good summary but lacks specific details about key findings. Including numerical results or specific genes/QTLs would enhance its informativeness.

The introduction is comprehensive but overly descriptive in some parts. Focus more on the knowledge gap and the study's unique contributions.

Include more references to recent studies on selection signals and genomic studies in sheep from diverse geographical contexts.

Describe the data quality control steps in more detail. For example, how were outlier SNPs or missing data handled?

Clarify the composite selection signal (CSS) methodology. Provide justification for using this approach over others (e.g., iHS, XP-EHH).

Were any statistical corrections applied to account for population structure or relatedness among individuals?

The presentation of the results is clear, but:

Include figures or tables summarizing key candidate genes or QTLs identified.

Highlight specific loci and their putative roles in traits relevant to Indian sheep (e.g., drought tolerance, disease resistance).

Consider including a heatmap or Manhattan plot for better visualization of selection signals.

The discussion lacks depth in interpreting biological functions of identified genes. Provide insights into how these genes/QTLs align with known traits in sheep.

Compare findings with results from other studies (e.g., on sheep from different regions or breeds).

Discuss potential limitations of the study, such as sample size or marker density, and how they might affect the robustness of the conclusions.

Figures are clear but lack detailed legends. Add more context to explain what the reader should infer from each figure.

Tables summarizing selection signals should include effect sizes and p-values where relevant.

**Reviewer 2**

Authors are requested to revise the manuscript as per the comments or suggestions put into the manuscript file uploaded in the system, also expected to address the issues related to Discussion section as under:

1. Link Between Genomic Regions and Local Adaptation: While the discussion mentions the environmental adaptations of the breeds, it could further elaborate on how these adaptations might have influenced the specific genomic regions under selection. For example, more direct comparisons to how the genetic variations might correlate with the climate or geographical features of the areas where these breeds are found could enhance the reader's understanding of the adaptive significance of these genes.

2. Consistency in Gene References: Some genes are mentioned repeatedly (e.g., CNTNAP5), but the context of how they were identified in different breeds could be more explicitly connected to avoid confusion. For instance, you might clarify how CNTNAP5's role in body growth is shared across breeds like CHA and GAR, yet each breed’s unique genomic context could influence the gene's effect.

3. Discussion on Statistical Methodology: The use of the DCMS method is discussed, but more detailed insight into why this method improves the resolution power of the genomic signals could help readers understand the methodological benefits. Briefly touching on the weaknesses of other statistical approaches and how the DCMS method overcomes them would strengthen the argument.

4. Gene Functionality and Mechanisms: While candidate genes are identified, more information on their molecular mechanisms or the processes through which they impact the phenotypic traits (e.g., how MTSS1 influences bacterial resistance) could help provide a clearer understanding of their biological relevance.

5. Comparison with Other Studies: The discussion of prior research, such as the studies from Ahmad et al. (2021) and Zhang et al. (2016), provides useful context. However, it would be helpful to briefly explain how the findings in the current study either align or contrast with these studies, reinforcing the study’s contribution to the field.

**Reviewer 3**

1. The manuscript, especially the discussion part is exceedingly long with redundancy in a few parts which is repeated in the results and discussion section.

2. Check the spelling “Chantangi” which is wrongly written.

3. Concise the discussion part and avoid redundancy.

Reviewers' comments:

Reviewer's Responses to Questions

**Comments to the Author**

1. Is the manuscript technically sound, and do the data support the conclusions?

Reviewer #1: Partly

Reviewer #2: Yes

Reviewer #3: Yes

2. Has the statistical analysis been performed appropriately and rigorously?

Reviewer #1: I Don't Know

Reviewer #2: Yes

Reviewer #3: Yes

3. Have the authors made all data underlying the findings in their manuscript fully available?

Reviewer #1: Yes

Reviewer #2: Yes

Reviewer #3: Yes

4. Is the manuscript presented in an intelligible fashion and written in standard English?

Reviewer #1: Yes

Reviewer #2: Yes

Reviewer #3: Yes

**Reviewer #1:** The study addresses an important topic in animal genetics, focusing on Indian sheep breeds, which are of economic and cultural importance. The identification of candidate genes and QTLs has significant implications for improving breed performance and conservation strategies.

However, the manuscript requires some clarifications and revisions to enhance its scientific rigor and readability.

The manuscript is well-organized overall, but some sections (e.g., methods and results) require more detailed explanations.

Certain paragraphs in the introduction and discussion are redundant and could be condensed.

Specific Comments:

The abstract provides a good summary but lacks specific details about key findings. Including numerical results or specific genes/QTLs would enhance its informativeness.

The introduction is comprehensive but overly descriptive in some parts. Focus more on the knowledge gap and the study's unique contributions.

Include more references to recent studies on selection signals and genomic studies in sheep from diverse geographical contexts.

Describe the data quality control steps in more detail. For example, how were outlier SNPs or missing data handled?

Clarify the composite selection signal (CSS) methodology. Provide justification for using this approach over others (e.g., iHS, XP-EHH).

Were any statistical corrections applied to account for population structure or relatedness among individuals?

The presentation of the results is clear, but:

Include figures or tables summarizing key candidate genes or QTLs identified.

Highlight specific loci and their putative roles in traits relevant to Indian sheep (e.g., drought tolerance, disease resistance).

Consider including a heatmap or Manhattan plot for better visualization of selection signals.

The discussion lacks depth in interpreting biological functions of identified genes. Provide insights into how these genes/QTLs align with known traits in sheep.

Compare findings with results from other studies (e.g., on sheep from different regions or breeds).

Discuss potential limitations of the study, such as sample size or marker density, and how they might affect the robustness of the conclusions.

Figures are clear but lack detailed legends. Add more context to explain what the reader should infer from each figure.

Tables summarizing selection signals should include effect sizes and p-values where relevant.

**Reviewer #2:** Authors are requested to revise the manuscript as per the comments or suggestions put into the manuscript file uploaded in the system, also expected to address the issues related to Discussion section as under:

1. Link Between Genomic Regions and Local Adaptation: While the discussion mentions the environmental adaptations of the breeds, it could further elaborate on how these adaptations might have influenced the specific genomic regions under selection. For example, more direct comparisons to how the genetic variations might correlate with the climate or geographical features of the areas where these breeds are found could enhance the reader's understanding of the adaptive significance of these genes.

2. Consistency in Gene References: Some genes are mentioned repeatedly (e.g., CNTNAP5), but the context of how they were identified in different breeds could be more explicitly connected to avoid confusion. For instance, you might clarify how CNTNAP5's role in body growth is shared across breeds like CHA and GAR, yet each breed’s unique genomic context could influence the gene's effect.

3. Discussion on Statistical Methodology: The use of the DCMS method is discussed, but more detailed insight into why this method improves the resolution power of the genomic signals could help readers understand the methodological benefits. Briefly touching on the weaknesses of other statistical approaches and how the DCMS method overcomes them would strengthen the argument.

4. Gene Functionality and Mechanisms: While candidate genes are identified, more information on their molecular mechanisms or the processes through which they impact the phenotypic traits (e.g., how MTSS1 influences bacterial resistance) could help provide a clearer understanding of their biological relevance.

5. Comparison with Other Studies: The discussion of prior research, such as the studies from Ahmad et al. (2021) and Zhang et al. (2016), provides useful context. However, it would be helpful to briefly explain how the findings in the current study either align or contrast with these studies, reinforcing the study’s contribution to the field.

**Reviewer #3:** 1. The manuscript, especially the discussion part is exceedingly long with redundancy in a few parts which is repeated in the results and discussion section.

2. Check the spelling “Chantangi” which is wrongly written.

3. Concise the discussion part and avoid redundancy.

**Do you want your identity to be public for this peer review?** For information about this choice, including consent withdrawal, please see our Privacy Policy

Reviewer #1: No

Reviewer #2: **Yes:** Dr. Ananta Kumar Das, Ph.D., Associate Professor, Department of Animal Genetics & Breeding, Faculty of Veterinary and Animal Sciences, WB University of Animal and Fishery Sciences, Kolkata-37 (IN)

Reviewer #3: **Yes:** Partha Pratim Das

---

## [Author Response · Author response to Decision Letter 1]

30 Mar 2025

Manuscript Number: PONE-D-24-51732

Title: Composite Selection Signal Analysis: Uncovering Candidate Genes and Quantitative Trait Loci in Indian Sheep Breeds

S.No. Reviewers Comments Response to Reviewer’s Comments

01 Certain paragraphs in the introduction and discussion are redundant and could be condensed. Thank you for pointing that out. The redundancy part in the discussion is removed

02 Specific Comments:

The abstract provides a good summary but lacks specific details about key findings. Including numerical results or specific genes/QTLs would enhance its informativeness. Thank you for bringing that issue to our attention. The main findings and significant numerical results have been clearly detailed in the abstract section.

03 The introduction is comprehensive but overly descriptive in some parts. Focus more on the knowledge gap and the study's unique contributions. I am grateful for your bringing this matter to my attention. I have incorporated a paragraph that addresses the existing knowledge gaps and underscores the distinctiveness of the current study.

04 Include more references to recent studies on selection signals and genomic studies in sheep from diverse geographical contexts. Thank you for your valuable input.

Recent research on selection signals and global studies on sheep have been included in the references.

05 Describe the data quality control steps in more detail. For example, how were outlier SNPs or missing data handled? Thank you for your comments about it. Detailed the steps for controlling data quality in the methods section.

06 Clarify the composite selection signal (CSS) methodology. Provide justification for using this approach over others (e.g., iHS, XP-EHH). I'm thankful for your notice on this matter. The methodology of composite selection signals and its advantages compared to alternative methods were incorporated.

07 Were any statistical corrections applied to account for population structure or relatedness among individuals? Thank you for your comment. Pi hat values are computed for all individuals in the study, and the genomic relatedness among them is low. As a result, all individuals are included in the selection signature analysis.

08 The presentation of the results is clear, but:

Include figures or tables summarizing key candidate genes or QTLs identified. I appreciate you bringing that to my attention. The text included a table that summarizes the important candidate genes.

09 Highlight specific loci and their putative roles in traits relevant to Indian sheep (e.g., drought tolerance, disease resistance). I'm grateful for your insight on this. The proposed genetic locations associated with traits such as drought tolerance and disease resistance were highlighted in the updated version.

10 Consider including a heatmap or Manhattan plot for better visualisation of selection signals. I appreciate you pointing that out to me. The manuscript now includes a Manhattan plot that effectively illustrates the selection signals identified in our analysis.

11 The discussion lacks depth in interpreting biological functions of identified genes. Thanks for shedding light on that. The specifics regarding the molecular functions of the proposed genes were included in the discussion section.

12 Provide insights into how these genes/QTLs align with known traits in sheep. Thanks for making me aware of it. The specified details have been incorporated into the manuscript in accordance with the recommendations provided.

13 Compare findings with results from other studies (e.g., on sheep from different regions or breeds). Thanks for the comment. Comparison of findings of this study with other studies were included in the revised manuscript.

14 Discuss potential limitations of the study, such as sample size or marker density, and how they might affect the robustness of the conclusions. I’m grateful for your comment. The potential limitations of the study are added in the manuscript.

15 Figures are clear but lack detailed legends. Add more context to explain what the reader should infer from each figure. Thank you for pointing out. However, the existing pie chart has the common legend, that explains the name of the QTL classes.

16 Tables summarizing selection signals should include effect sizes and p-values where relevant Thanks for the comment. Changes incorporated as per the comment.

Manuscript Number: PONE-D-24-51732

Title:Title: Composite Selection Signal Analysis: Uncovering Candidate Genes and Quantitative Trait Loci in Indian Sheep Breeds

Reviewer 02

Specific Comments

S.No. Reviewer’s Comments Response

01 Link Between Genomic Regions and Local Adaptation: While the discussion mentions the environmental adaptations of the breeds, it could further elaborate on how these adaptations might have influenced the specific genomic regions under selection. For example, more direct comparisons to how the genetic variations might correlate with the climate or geographical features of the areas where these breeds are found could enhance the reader's understanding of the adaptive significance of these genes. Thank you for the suggestion. Incorporated the text as suggested

02 Consistency in Gene References: Some genes are mentioned repeatedly (e.g., CNTNAP5), but the context of how they were identified in different breeds could be more explicitly connected to avoid confusion. For instance, you might clarify how CNTNAP5's role in body growth is shared across breeds like CHA and GAR, yet each breed’s unique genomic context could influence the gene's effect. I’m grateful for your suggestion and have integrated the text as suggested.

03 Discussion on Statistical Methodology: The use of the DCMS method is discussed, but more detailed insight into why this method improves the resolution power of the genomic signals could help readers understand the methodological benefits. Briefly touching on the weaknesses of other statistical approaches and how the DCMS method overcomes them would strengthen the argument Thanks for the recommendation; I’ve added the text accordingly.

04 Gene Functionality and Mechanisms: While candidate genes are identified, more information on their molecular mechanisms or the processes through which they impact the phenotypic traits (e.g., how MTSS1 influences bacterial resistance) could help provide a clearer understanding of their biological relevance I am thankful for your suggestion and have made the changes to the text as proposed.

05 Comparison with Other Studies: The discussion of prior research, such as the studies from Ahmad et al. (2021) and Zhang et al. (2016), provides useful context. However, it would be helpful to briefly explain how the findings in the current study either align or contrast with these studies, reinforcing the study’s contribution to the field I appreciate your suggestion and have appended the changes to the text as proposed.

General Comments

01 Rephrase as "False Discovery Rate (FDR) threshold of < 5%" Modified as suggested

02 Genes could be presented by trait type e.g., morphology (putative genes with sub-trait), growth (putative genes with sub-trait), reproduction (putative genes with sub-trait), adaptation (putative genes with sub-trait), disease resistance (putative genes with sub-trait), etc. Changes made as per the correction- Genes rearranged into sub-traits.

03 QTLs could be grouped by trait also. The QTLs were grouped according to the the trait - modified as per the recommendations

04 Referencing [1] is just akwarding as the sentence approaches to introduce a concept and hence remove the reference Removed the reference as suggested

05 Explain why the concept of selection signatures is important or how understanding them benefits evolutionary biology. Updated as advised

06 The phrase "mutations" seems redundant, as mutations are already mentioned in the previous sentence, so restructure the present sentence. Restructure the sentence as recommended

07 This is clear but could be more engaging. Consider adding a sentence or two on how SNP arrays have specifically impacted genomic studies, perhaps by providing concrete examples of successful applications. Enhanced as proposed

08 Give slight details about XPEHH Refined based on your feedback

09 How does DCMS method work or why it seems more robust and sensitive? Edited in line with your comments.

10 This sentence seems to appear abruptly. A sentence or two linking the SNP arrays discussed previously to the OvineSNP50 Bead Chip would provide a smoother transition. Revised according to your recommendation

11 Authors might add a bit more context about why the OvineSNP50 is specifically chosen for Indian sheep genome studies. For example, is it due to its size, variety of SNPs, or its validation in Indian sheep breeds? Updated as advised

12 Authors might use terms like "cold arid regions of the Western Himalayas" or "semi-arid regions of the Deccan Plateau", etc. for clarity Adjusted as indicated

13 Correct spelling (Odisha) Corrected as suggested

14 This is a strong statement that highlights the gap in research. Authors could emphasize why this gap is significant, perhaps briefly mentioning how genomics could benefit sheep production in India. Amended as suggested

15 Rephrase for clarity, such as: "In this study, we applied the composite signals method (DCMS) to detect selection signatures in the genomes of three Indian sheep breeds (Chantangi, Deccani, and Garole), which are adapted to diverse ecological niches and production systems across India. Changed as per your suggestion.

16 Briefly mention why these particular breeds are chosen. Are they important economically, or do they exhibit distinct phenotypic traits? Updated following your suggestion

17 It will be better to represent in a single sentence highlighting the goat or objective of the study. Amended in accordance with your feedback

18 Rephrase for better flow: "In the present study, we analyzed high-density genotypes from three Indian sheep breeds: Changthangi (CHA, n = 29), Deccani (IDC, n = 24), and Garole (GAR, n = 26), genotyped using the Illumina Ovine SNP50 BeadChip, which were obtained from WIDDE [14]." Refined as you advised

19 Consider providing a bit more context about WIDDE (e.g., what it stands for or its relevance) for clarity. Enhanced based on your thoughts

20 For clarity, authors might say: "The Illumina Ovine SNP50 BeadChip consists of 54,241 SNPs distributed across the sheep genome, with an average density of one SNP every 51 kb." Rephrased as recommended

21 Improve for clarity and consistency: "Quality control (QC) of the genotypes was performed based on the following criteria: minor allele frequency (maf < 0.05), Hardy-Weinberg equilibrium (HWE p < 0.0001), individuals with missing genotypes (--mind < 0.1), and SNPs with missing genotype data (geno < 0.1)." Updated following your suggestion

22 It would be helpful to clarify why these specific thresholds (maf, HWE, etc.) were chosen.

23 Authors might briefly explain why X-chromosome SNPs were excluded, especially if it's relevant to the study design (e.g., if focusing only on autosomes). Explained following your feedback

24 Authors could expand this purpose of PCA slightly on what they hoped to achieve with PCA (e.g., confirming population structure or identifying outliers). Expanded as suggested

25 Mention briefly what the "snprelate" package does. For example, authors could say: "The PCA was performed using the 'snprelate' package in R, which is designed for efficient analysis of large-scale genotypic data." Refined as you advised.

26 Authors might also consider adding a sentence explaining why DCMS is preferred in this context or how it enhances the analysis of selection signals. Enhanced as proposed

27 The present study utilized the Composite of Multiple Signals (DCMS) method, as detailed in Yurchenko et al. (2018), combining five univariate statistics: FST, Haplotype Homozygosity (H1), Modified Haplotype Homozygosity (H12), Tajima’s D, and Nucleotide Diversity (π) [18–21]." Corrected following your input

28 Correct as 'locus' Edited as per your suggestion

29 Consider adding a more intuitive explanation of why these components are combined (e.g., "The DCMS statistic combines both the p-values of individual statistics and their correlations to improve the sensitivity and robustness of detecting selection signals."). Refined based on your feedback

30 briefly mention why FST is used (e.g., to quantify genetic differentiation). Updated following your suggestion

31 explain a bit more about why these statistics are important and how they contribute to detecting selection signals (e.g., "Haplotype homozygosity measures the extent to which haplotypes are conserved, which can help identify regions under selective pressure."). Explained, following your advice

32 consider elaborating on how they relate to detecting selection. For example, "Tajima's D detects departures from neutrality, with negative values indicating potential selective sweeps, while nucleotide diversity (π) measures the genetic variation within a population." Enhanced as proposed

33 Authors might rephrase as "The MINOTAUR package was used to compute p-values for the five statistics based on their functional ranks, integrating multiple signals into a composite measure." Altered as per the comment

34 Consider adding a sentence or two that briefly explains the role of these steps in improving the accuracy of the results. Adapted per your suggestion

35 Modify the criteria for determining putative regions as "A q-value threshold of 0.05 was chosen to identify genomic regions with strong evidence of selection, while adjacent SNPs with q-values greater than 0.1 were excluded to reduce false positives."). Changed in response to your advice

36 clarify how the QTL enrichment analysis is performed. Authors might write as "QTL enrichment analysis was performed to identify overrepresented traits within the significant regions, using a genome-wide approach and a false discovery rate (FDR) threshold of 0.05."). Revised based on your input

37 The terms are essentially redundant. Consider: "missingness thresholds for SNP genotypes (geno <0.1) and individual genotypes (mind <0.1)." Altered per your guidance

38 Exact number or percentages of SNPs excluded for each criterion are missing. Included in accordance with your feedback

39 Exact number or percentage of SNPs excluded for this criterion is missing. Providing these numbers or percentages of all criteria would add more transparency to the data quality process. Included in accordance with your feedback

40 Modify as "The PCA confirmed the genetic differentiation among the three breeds, as indicated by their clear separation into distinct clusters. Corrected following your input

41 May be presented as "PC1 accounted for 14.34% of the variation, highlighting the strongest genetic differences between the populations, while PC2 explained an additional 7.98% of the total variation, capturing finer distinctions." Refined as you advised

42 correct whether it is "Chantangi" or "Changthangi." Corrected as per your suggestion

43 The size range for genomic regions under selection (from 1080.32 to 1.83 Kb) is inconsistent with the typical size of genomic regions under selection in animal breeding studies. Consider reviewing these figures (DCMS Metrics) for accuracy. We agreed to the statement you pointed out. However, the average length of genomic regions under selection was consistent with the literature, but not range. In our study, the range of genomic size we observed was not consistent with the literature.

44 It would help to summarize the QTL classes before delving into breed-specific information by adding one sentence like "The QTLs identified in each breed were categorized into various trait classes, including body weight, reproduction, milk yield, and health." Added a sentence following your suggestion

45 but the results could be more digestible if key QTLs from the tables were also summarized in the text. For example, "The most significant QTL associated with milk yield in CHA sheep was found on OAR 16, as shown in Table 3." Modified the sentence as per your feedback

46 Recheck to ensure the consistency and biological relevance of the findings.

---

## [Decision Letter · Decision Letter 1]

20 Apr 2025

Dear Dr. Destaw Worku Worku,

Thank you for submitting your manuscript to PLOS ONE. After careful consideration, we feel that it has merit but does not fully meet PLOS ONE’s publication criteria as it currently stands. Therefore, we invite you to submit a revised version of the manuscript that addresses the points raised during the review process.

Dear Dr., Destaw Worku Worku

Thank you for submitting your manuscript to PLOS ONE. After careful consideration, we have decided that your manuscript needs Minor Revision.

Kind regards,

Prof. Lamiaa Mostafa Radwan, Ph.D.

Academic Editor

PLOS ONE

**Reviewer 2**

Authors are requested to adhere the comments/ suggestion/corrections inscribed in the attached manuscript pdf file for thorough revision satisfying journal's standard presentation style along with reference checking, typographical and grammatical corrections.

**Reviewer 3**

The authors have addressed the reviewers' comments elaborately. This study highlights important findings on QTL, which encompass significant protein-coding gene traits in Indian indigenous sheep breeds of economic interest. The study presents findings that could have a significant impact on the scientific community in time.

We look forward to receiving your revised manuscript.

Kind regards,

Lamiaa Mostafa Radwan, Ph.D.

Academic Editor

PLOS ONE

Journal Requirements:

Additional Editor Comments (if provided):

Dear Dr., Destaw Worku Worku

Thank you for submitting your manuscript to PLOS ONE. After careful consideration, we have decided that your manuscript needs Minor Revision.

Kind regards,

Prof. Lamiaa Mostafa Radwan, Ph.D.

Academic Editor

PLOS ONE

Reviewer 2

Authors are requested to adhere the comments/ suggestion/corrections inscribed in the attached manuscript pdf file for thorough revision satisfying journal's standard presentation style along with reference checking, typographical and grammatical corrections.

Reviewer 3

The authors have addressed the reviewers' comments elaborately. This study highlights important findings on QTL, which encompass significant protein-coding gene traits in Indian indigenous sheep breeds of economic interest. The study presents findings that could have a significant impact on the scientific community in time.

Reviewers' comments:

Reviewer's Responses to Questions

**Comments to the Author**

Reviewer #2: All comments have been addressed

Reviewer #3: All comments have been addressed

2. Is the manuscript technically sound, and do the data support the conclusions?

Reviewer #2: Yes

Reviewer #3: Partly

3. Has the statistical analysis been performed appropriately and rigorously?

Reviewer #2: Yes

Reviewer #3: Yes

4. Have the authors made all data underlying the findings in their manuscript fully available?

Reviewer #2: Yes

Reviewer #3: Yes

5. Is the manuscript presented in an intelligible fashion and written in standard English?

Reviewer #2: No

Reviewer #3: Yes

Reviewer #2: Authors are requested to adhere the comments/ suggestion/corrections inscribed in the attached manuscript pdf file for thorough revision satisfying journal's standard presentation style along with reference checking, typographical and grammatical corrections.

Reviewer #3: The authors have addressed the reviewers' comments elaborately. This study highlights important findings on QTL, which encompass significant protein-coding gene traits in Indian indigenous sheep breeds of economic interest. The study presents findings that could have a significant impact on the scientific community in time.

**Do you want your identity to be public for this peer review?** For information about this choice, including consent withdrawal, please see our Privacy Policy

Reviewer #2: **Yes:** Dr. Ananta Kumar Das, Associate Professor, Department of Animal Genetics and Breeding, West Bengal University of Animal and Fishery Sciences, Kolkata (IN)

Reviewer #3: No

---

## [Author Response · Author response to Decision Letter 2]

3 May 2025

S. No. Journal requirements

01. Line number 83: Follow authors' instruction for citing reference(s); here needs reference number in parenthesis i.e. [10]

Response: The authors thank you for pointing that out. We followed the authors' instructions in citing the references

02. Line number 93: Follow authors' instruction for citing reference(s); here needs reference numbers in parentheses i.e. [.....] and [.....]

Response: Thank you for pointing that out. We followed the author’s instructions in citing the references.

03 Line number 103-104: Rephrase the sentence as “including few Indian sheep breeds.”

Response: Rephrased the sentence as per the comment.

04 Line number 109: Give data as per the 20th census

Response: The sheep population, according to the 20th livestock census, is provided.

05 Line number 124: Follow authors' instruction for citing reference(s); here needs reference number in parenthesis i.e. [........]

Response: Thank you for pointing that out. We followed the author’s instructions in citing the references.

06 Line number 156: Change De-Correlated Composite of Multiple Selection Signals to DCMS.

Response: Correction made according to the comment.

07 Line number 157: Change DCMS to ‘The DCMS’

Response: Revision is made as per the comment.

08 Line number 158: Follow authors' instruction for citing reference(s); here needs reference number in parenthesis i.e. [.......]

Response: Thank you for pointing that out, sir. We followed the author’s instructions in citing the references.

09 Line number 160: Change FST to Fixation index (FST)

Response: Revised the name of the statistic.

10 Line number 161: Change Tajima’s D to Tajima D index.

Response: Corrected the name of the statistic.

11 Line number 162: Shift into the space just before line 162.

Response: Corrected the name of the statistic.

12 Line number 178: format in subscript Fst.

Response: Formatted according to the comment.

13 Line number 196: Change the sentence to “These statistics”.

Response: Revised the sentence according to the comment.

14 Line number 202: Delete the sentence “as outlined in”

Response: Deleted the redundant sentence

15 Line number 205: use 'statistics' only because statistics are specific to sample estimates and parameters are specific to population estimates.

Response: The word statistic is used instead of statistic parameters as suggested.

16 Line number 213: Cite reference number in parenthesis [.......]

Response: Thank you for pointing that out, sir. We followed the author’s instructions in citing the references.

17 Line number 214: ???

Response: Needful done

18 Line number 234: close the bracket

Response: Changes were made as per the correction

19 Line number 251: crosscheck with the tabulated data

Response: The correct values are now added after cross-check with the data in the table

20 Line number 282: QTL associated with the total....

Response: Changes made as per the suggestion.

21 Line number 321: Delete “has”

Response: According to the correction, the word ‘has’ is deleted.

22 Line number 341: Delete “has”; delete and follow the simple past tense throughout the Results.

Response: According to the correction, the word ‘has’ is deleted, and throughout the results section, the simple past tense is used.

23 Line number 378: check formatting

Response: The changes have been made as suggested.

24 Line number 395: Cite reference number in parenthesis [.......]

Response: Thank you for pointing that out, sir. We followed the author’s instructions in citing the references.

25 Line number 413: check formatting

Response: Needful done

26 Line number 490: Cite reference number in parenthesis [.......]

Response: Thank you for pointing that out, sir. We followed the author’s instructions in citing the references.

27 Line number 497: Cite reference number in parenthesis [.......]

Response: Thank you for pointing that out, sir. We followed the author’s instructions in citing the references.

28 Line number 502: Cite reference number in parenthesis [.......]

Response: Thank you for pointing that out, sir. We followed the author’s instructions in citing the references.

29 Line number 514: Cite reference number in parenthesis [.......]

Response: Thank you for pointing that out, sir. We followed the author’s instructions in citing the references.

30 Line number 604: Cite reference number in parenthesis [.......]

Response: Needful done

31 Line number 655: delete and shift to line 656.

Response: The sentence at line 649 is removed and shifted to line number 656 as per the suggestion.

32 Line 657-658: Delete lines 651 and 652.

Response: The lines at 651 and 652 are deleted as per the comment.

---

## [Decision Letter · Decision Letter 2]

3 Jul 2025

Dear Dr. Worku,

<o:p></o:p>

Thank you for submitting your manuscript to PLOS ONE. After careful consideration, we feel that it has merit but does not fully meet PLOS ONE’s publication criteria as it currently stands. Therefore, we invite you to submit a revised version of the manuscript that addresses the points raised during the review process.

Dear Dr., Destaw Worku Worku

Thank you for submitting your manuscript to PLOS ONE. After careful consideration, we have decided that your manuscript needs Minor Revision.

Kind regards,

Prof. Lamiaa Mostafa Radwan, Ph.D.

Academic Editor

PLOS ONE

**Reviewer 1**

This study applied the De-Correlated Composite of Multiple Selection Signals (DCMS) approach to systematically analyze genomic selection signatures in three Indian sheep breeds: Chanthangi, Garole, and Deccani. Candidate genes and QTLs associated with key traits—including growth, reproduction, disease resistance, and environmental adaptation—were successfully identified, offering novel insights into the molecular mechanisms underlying genetic improvement in Indian sheep. However, several questions remain as follows:

1.The manuscript mentions that DCMS combines five statistics including FST, H1, H12, Tajima’s D, and π, but does not elaborate on the rationale for selecting this specific combination of five indicators. Have other statistics (such as iHS and XP-EHH) been evaluated for their complementarity? How are the weights of each statistic determined in the composite calculation?

2.Line81, It is recommended to add literature citations, such as doi: 10.1038/s41467-020-16485-1, doi: 10.24272/j.issn.2095-8137.2022.124.

3. Line 130, why choose these three breeds for the research?

4. Line225, please remove the colon

5. It is recommended to adjust the format of the manuscript, such as Table 8.

6. Has the QTL identified in this study overlapped with those previously reported?

We look forward to receiving your revised manuscript.

Kind regards,

Lamiaa Mostafa Radwan, Ph.D.

Academic Editor

PLOS ONE

Journal Requirements:

Additional Editor Comments:

Dear Dr., Destaw Worku Worku

Thank you for submitting your manuscript to PLOS ONE. After careful consideration, we have decided that your manuscript needs Minor Revision.

Kind regards,

Prof. Lamiaa Mostafa Radwan, Ph.D.

Academic Editor

PLOS ONE

Reviewer 1

This study applied the De-Correlated Composite of Multiple Selection Signals (DCMS) approach to systematically analyze genomic selection signatures in three Indian sheep breeds: Chanthangi, Garole, and Deccani. Candidate genes and QTLs associated with key traits—including growth, reproduction, disease resistance, and environmental adaptation—were successfully identified, offering novel insights into the molecular mechanisms underlying genetic improvement in Indian sheep. However, several questions remain as follows:

1.The manuscript mentions that DCMS combines five statistics including FST, H1, H12, Tajima’s D, and π, but does not elaborate on the rationale for selecting this specific combination of five indicators. Have other statistics (such as iHS and XP-EHH) been evaluated for their complementarity? How are the weights of each statistic determined in the composite calculation?

2.Line81, It is recommended to add literature citations, such as doi: 10.1038/s41467-020-16485-1, doi: 10.24272/j.issn.2095-8137.2022.124.

3. Line 130, why choose these three breeds for the research?

4. Line225, please remove the colon

5. It is recommended to adjust the format of the manuscript, such as Table 8.

6. Has the QTL identified in this study overlapped with those previously reported?

Reviewers' comments:

Reviewer's Responses to Questions

**Comments to the Author**

Reviewer #2: All comments have been addressed

Reviewer #4: (No Response)

2. Is the manuscript technically sound, and do the data support the conclusions?

Reviewer #2: Yes

Reviewer #4: (No Response)

3. Has the statistical analysis been performed appropriately and rigorously?

Reviewer #2: Yes

Reviewer #4: (No Response)

4. Have the authors made all data underlying the findings in their manuscript fully available?

Reviewer #2: Yes

Reviewer #4: (No Response)

5. Is the manuscript presented in an intelligible fashion and written in standard English?

Reviewer #2: Yes

Reviewer #4: (No Response)

Reviewer #2: (No Response)

Reviewer #4: This study applied the De-Correlated Composite of Multiple Selection Signals (DCMS) approach to systematically analyze genomic selection signatures in three Indian sheep breeds: Chanthangi, Garole, and Deccani. Candidate genes and QTLs associated with key traits—including growth, reproduction, disease resistance, and environmental adaptation—were successfully identified, offering novel insights into the molecular mechanisms underlying genetic improvement in Indian sheep. However, several questions remain as follows:

1.The manuscript mentions that DCMS combines five statistics including FST, H1, H12, Tajima’s D, and π, but does not elaborate on the rationale for selecting this specific combination of five indicators. Have other statistics (such as iHS and XP-EHH) been evaluated for their complementarity? How are the weights of each statistic determined in the composite calculation?

2.Line81, It is recommended to add literature citations, such as doi: 10.1038/s41467-020-16485-1, doi: 10.24272/j.issn.2095-8137.2022.124.

3. Line 130, why choose these three breeds for the research?

4. Line225, please remove the colon

5. It is recommended to adjust the format of the manuscript, such as Table 8.

6. Has the QTL identified in this study overlapped with those previously reported?

**Do you want your identity to be public for this peer review?** For information about this choice, including consent withdrawal, please see our Privacy Policy

Reviewer #2: **Yes:** Prof. (Dr.) Ananta Kumar Das, Department of Animal Genetics and Breeding, Faculty of Veterinary and Animal Sciences, West Bengal University of Animal and Fishery Sciences, Kolkata-37 (IN)

Reviewer #4: No

---

## [Author Response · Author response to Decision Letter 3]

2 Aug 2025

Manuscript Number: PONE-D-24-51732R2

Title: Composite Selection Signal Analysis: Uncovering Candidate Genes and Quantitative Trait Loci in Indian Sheep Breeds

S.No. Reviewer’s comments Response to Comments

01. The manuscript mentions that DCMS combines five statistics including FST, H1, H12, Tajima’s D, and π, but does not elaborate on the rationale for selecting this specific combination of five indicators. Have other statistics (such as iHS and XP-EHH) been evaluated for their complementarity? How are the weights of each statistic determined in the composite calculation? • Thank you, Sir, for pointing this out. The rationale for selecting this specific set of five indicators in the study arises from the need to assess genomic selection signatures through combining the selection signals obtained from the univariate statistics. We utilised FST to evaluate population differentiation, while h1 and h12 were employed for identifying haplotype-based signatures based on linkage disequilibrium. To capture signals related to the allele spectrum, we applied Tajima’s D and nucleotide diversity (π). The same is incorporated at the line numbers from 89-93.

• Further, h12 is a comparable statistic to integrated Haplotype Score (iHS), and XP-EHH is also utilized for examining between-breed signatures only. This study is mainly based on within breed signatures. Hence, XP-EHH is not used in this study.

• We obtained the q value (combined) by the following steps:

1. For each statistic within a given breed, genome-wide p-values based on fractional ranks (i.e. the stat_to_pvalue function) of the Minotour R package

2. Left tailed and right tailed tests were performed on the statistic, based on the distributions.

3. n x n correlation matrix among the statistic (minimum covariance determinant estimator of the multivariate location and scatter) using the covNAMcd function of the Minotour R package. Then, DCMS values are generated.

4. The DCMS values are fitted to a normal distribution using the MASS package in R.

5. Then p-values are calculated from the DCMS values using the pnorm function in the MASS package.

6. Finally, the p-values are converted to the q-values using the Benjamini and Hochberg method using the p.adjust R function.

7. The DCMS calculation employed in this study is based on the Yurchenko et al., 2018, which is further based on Lotterhos et al., 2017.

8. The DCMS values are mainly obtained by the nxn correlation matrix, which is obtained due to the covariance existed among the univariate statistics.

9. This method is more efficient and different from the method proposed by Randhawa et al., 2015. The latter study is based on the calculation of weights for the univariate statistic.

02. Line81, It is recommended to add literature citations, such as doi: 10.1038/s41467-020-16485-1, doi: 10.24272/j.issn.2095-8137.2022.124. Thank you for your suggestion, the recommended references were added at line 80.

03 Line 130, why choose these three breeds for the research? In this research, we focused on three indigenous sheep breeds—Changtangi, Garole, and Deccani—selected for their unique agroclimatic adaptations and production attributes. The Changtangi breed grows in the frigid, high-altitude areas of the Northern Himalayas, while Garole is adapted to hot and humid climates, and Deccani thrives in the arid, dry regions of India. This deliberate selection allows for the identification of quantitative trait loci (QTLs) linked to adaptation to thermal and cold stress across varying environmental conditions. Moreover, these breeds collectively encompass a diverse range of production traits: Changtangi is valued for its fine wool, Deccani is preferred for meat production, and Garole is noted for its prolific reproductive capabilities. Incorporating these breeds into our study significantly broadens the investigation into the genetic architecture that governs environmental adaptability and economically relevant traits in Indian sheep.

04 Line225, please remove the colon Thank you for the comment. The colon was removed

05 It is recommended to adjust the format of the manuscript, such as Table 8. Thank you for the comment. The format of the Table 8 was adjusted

06 Has the QTL identified in this study overlapped with those previously reported? In this study, we observed a minimal overlap with the findings of the other studies such as Ahmad et al., 2021 and Sarvanan et al., 2021.

---

## [Decision Letter · Decision Letter 3]

21 Aug 2025

Dear Dr. Worku,

Thank you for submitting your manuscript to PLOS ONE. After careful consideration, we feel that it has merit but does not fully meet PLOS ONE’s publication criteria as it currently stands. Therefore, we invite you to submit a revised version of the manuscript that addresses the points raised during the review process.

Dear Dr. Destaw Worku Worku

Thank you for submitting your manuscript to PLOS ONE. After careful consideration, we have decided that your manuscript needs Mainor Revision.

Kind regards,

Prof. Lamiaa Mostafa Radwan, Ph.D.

Academic Editor

PLOS ONE

**Reviewer3**

The manuscript has undergone slight modifications. However, there are numerous grammatical errors. In several instances, a sentence is written in the past tense, while the following one is in the future tense. For consistency, it is recommended that the Methods section be written entirely in the past tense and maintain uniformity. A thorough review of the manuscript’s grammar is also recommended and avoid overall redundancy.

In line 395, the phrase “researchers could not differentiate” is unclear. It is indistinct whether the author is referring to findings from the previously published paper by Ahmed et al. or to their own data. Recommended to rephrase this sentence to clearly convey the intended meaning.

**Reviewer4**

The authors have comprehensively addressed the concerns and suggestions from the initial review. The revised manuscript demonstrates notable improvements. Therefore, the manuscript can be accepted.

We look forward to receiving your revised manuscript.

Kind regards,

Lamiaa Mostafa Radwan, Ph.D.

Academic Editor

PLOS ONE

Journal Requirements:

Additional Editor Comments :

Dear Dr. Destaw Worku Worku

Thank you for submitting your manuscript to PLOS ONE. After careful consideration, we have decided that your manuscript needs Mainor Revision.

Kind regards,

Prof. Lamiaa Mostafa Radwan, Ph.D.

Academic Editor

PLOS ONE

Reviewer3

The manuscript has undergone slight modifications. However, there are numerous grammatical errors. In several instances, a sentence is written in the past tense, while the following one is in the future tense. For consistency, it is recommended that the Methods section be written entirely in the past tense and maintain uniformity. A thorough review of the manuscript’s grammar is also recommended and avoid overall redundancy.

In line 395, the phrase “researchers could not differentiate” is unclear. It is indistinct whether the author is referring to findings from the previously published paper by Ahmed et al. or to their own data. Recommended to rephrase this sentence to clearly convey the intended meaning.

Reviewer4

The authors have comprehensively addressed the concerns and suggestions from the initial review. The revised manuscript demonstrates notable improvements. Therefore, the manuscript can be accepted.

Reviewers' comments:

Reviewer's Responses to Questions

**Comments to the Author**

Reviewer #3: All comments have been addressed

Reviewer #4: All comments have been addressed

2. Is the manuscript technically sound, and do the data support the conclusions?

Reviewer #3: Yes

Reviewer #4: Yes

3. Has the statistical analysis been performed appropriately and rigorously?

Reviewer #3: Yes

Reviewer #4: Yes

4. Have the authors made all data underlying the findings in their manuscript fully available?

Reviewer #3: Yes

Reviewer #4: Yes

5. Is the manuscript presented in an intelligible fashion and written in standard English?

Reviewer #3: No

Reviewer #4: Yes

Reviewer #3: The manuscript has undergone slight modifications. However, there are numerous grammatical errors. In several instances, a sentence is written in the past tense, while the following one is in the future tense. For consistency, it is recommended that the Methods section be written entirely in the past tense and maintain uniformity. A thorough review of the manuscript’s grammar is also recommended and avoid overall redundancy.

In line 395, the phrase “researchers could not differentiate” is unclear. It is indistinct whether the author is referring to findings from the previously published paper by Ahmed et al. or to their own data. Recommended to rephrase this sentence to clearly convey the intended meaning.

Reviewer #4: The authors have comprehensively addressed the concerns and suggestions from the initial review. The revised manuscript demonstrates notable improvements. Therefore, the manuscript can be accepted.

**Do you want your identity to be public for this peer review?** For information about this choice, including consent withdrawal, please see our Privacy Policy

Reviewer #3: No

Reviewer #4: No

---

## [Author Response · Author response to Decision Letter 4]

23 Sep 2025

We sincerely thank the reviewer for their insightful comments and suggestions regarding our manuscript titled “Composite Selection Signal Analysis: Uncovering Candidate Genes and Quantitative Trait Loci in Indian Sheep Breeds” (Manuscript ID: PONE-D-24-51732R3). We have carefully revised the manuscript to address the concerns raised. Below are our detailed responses to each comment.

Comment 1:

The manuscript has undergone slight modifications. However, there are numerous grammatical errors. In several instances, a sentence is written in the past tense, while the following one is in the future tense. For consistency, it is recommended that the Methods section be written entirely in the past tense and maintain uniformity. A thorough review of the manuscript’s grammar is also recommended to avoid overall redundancy.

Response:

We appreciate the reviewer’s careful reading and valuable feedback. In response, we have conducted a comprehensive grammatical review of the entire manuscript. Particular attention was paid to tense consistency, especially in the Methods section, which has now been revised to use the past tense uniformly. We have also removed redundant phrases and improved sentence structure throughout the manuscript to enhance clarity and readability.

Reviewer Comment 2:

In line 395, the phrase “researchers could not differentiate” is unclear. It is indistinct whether the author is referring to findings from the previously published paper by Ahmed et al. or to their own data. Recommended to rephrase this sentence to clearly convey the intended meaning.

Response:

Thank you for pointing out this ambiguity. We have revised the sentence to clarify the reference as suggested.

During the revision process, we became aware of the recent publication of a related article, “Comparative genomic insights into adaptation, selection signatures, and population dynamics in indigenous Indian sheep and foreign breeds” (Frontiers in Genetics, August 2025, Muthusamy et al.). We have cited this paper in our revised manuscript and clearly discussed both the similarities and the novel aspects of our work. Our study differs from theirs in analytical approach, breed focus, and resolution, as well as the discovery of several novel candidate genes and QTLs specific to Indian sheep breeds.

We hope these revisions adequately address the reviewer’s concerns. We are grateful for the opportunity to improve our manuscript and thank you for your continued consideration.

Destaw Worku, PhD

Corresponding Author

---

## [Decision Letter · Decision Letter 4]

16 Oct 2025

Dear Dr. Worku,

Thank you for submitting your manuscript to PLOS ONE. After careful consideration, we feel that it has merit but does not fully meet PLOS ONE’s publication criteria as it currently stands. Therefore, we invite you to submit a revised version of the manuscript that addresses the points raised during the review process.

Dear Dr.  Destaw Worku

Thank you for submitting your manuscript to PLOS ONE. After careful consideration, we have decided that your manuscript needs Minor Revision.

Kind regards,

Prof. Lamiaa Mostafa Radwan, Ph.D.

Academic Editor

PLOS ONE

**Reviewer 3**

The authors has nicely addressed all the comments raised by reviewer. This study highlights important findings on QTL, which encompass significant protein-coding gene traits in Indian indigenous sheep breeds of economic interest. The study presents findings that could have a significant impact on the scientific community in time

**Reviewer 5**

General Comment: rewrite the entire document.

Abstract: Modify the statements such as “line 32: Natural and artificial selection shape the genomes of sheep”, “line 34: to preserve and enhance native genetic traits”, and “line 48: wide variety of characteristics”. The way of expression is not professional.

1. Give a description for each breed in the introduction section.

2. Rewrite from line: 87-90

3. A quick transition disrupts the flow of idea between paragraphs 2 and 3.

4. Line 118: “highly dense genotypes”. What does it mean? Please be sure to express the statement in professional terms.

5. Modify as 2.1 Ethics Statement and 2.2 Animal Selection and Genotyping

6. Line 246-247: 3.3.1

7. Line 282: production (9.38%). Which type of production?

8. Table 1 is not under the appropriate position because it is a common table for all breeds. You have already written the information of table 1 in text. No need of duplication.

9. In the discussion section try to elaborate on the general implication of your study, any limitation, and agreeing or disagreeing with previous reports with citations. No need of repeating the methodology or introductory information.

10. Captions of figures should be self-explanatory. So, explain each graph in detail.

We look forward to receiving your revised manuscript.

Kind regards,

Lamiaa Mostafa Radwan, Ph.D.

Academic Editor

PLOS ONE

**Journal Requirements:**

**Additional Editor Comments:**

Dear Dr. Destaw Worku

Thank you for submitting your manuscript to PLOS ONE. After careful consideration, we have decided that your manuscript needs Minor Revision.

Kind regards,

Prof. Lamiaa Mostafa Radwan, Ph.D.

Academic Editor

PLOS ONE

**Reviewer 3**

The authors has nicely addressed all the comments raised by reviewer. This study highlights important findings on QTL, which encompass significant protein-coding gene traits in Indian indigenous sheep breeds of economic interest. The study presents findings that could have a significant impact on the scientific community in time

**Reviewer 5**

General Comment: rewrite the entire document.

Abstract: Modify the statements such as “line 32: Natural and artificial selection shape the genomes of sheep”, “line 34: to preserve and enhance native genetic traits”, and “line 48: wide variety of characteristics”. The way of expression is not professional.

1. Give a description for each breed in the introduction section.

2. Rewrite from line: 87-90

3. A quick transition disrupts the flow of idea between paragraphs 2 and 3.

4. Line 118: “highly dense genotypes”. What does it mean? Please be sure to express the statement in professional terms.

5. Modify as 2.1 Ethics Statement and 2.2 Animal Selection and Genotyping

6. Line 246-247: 3.3.1

7. Line 282: production (9.38%). Which type of production?

8. Table 1 is not under the appropriate position because it is a common table for all breeds. You have already written the information of table 1 in text. No need of duplication.

9. In the discussion section try to elaborate on the general implication of your study, any limitation, and agreeing or disagreeing with previous reports with citations. No need of repeating the methodology or introductory information.

10. Captions of figures should be self-explanatory. So, explain each graph in detail.

Reviewers' comments:

Reviewer's Responses to Questions

**Comments to the Author**

Reviewer #3: All comments have been addressed

Reviewer #5: (No Response)

2. Is the manuscript technically sound, and do the data support the conclusions?

Reviewer #3: Yes

Reviewer #5: Yes

3. Has the statistical analysis been performed appropriately and rigorously?

Reviewer #3: Yes

Reviewer #5: Yes

4. Have the authors made all data underlying the findings in their manuscript fully available?

Reviewer #3: Yes

Reviewer #5: Yes

5. Is the manuscript presented in an intelligible fashion and written in standard English?

Reviewer #3: Yes

Reviewer #5: No

**Reviewer #3:**  The authors has nicely addressed all the comments raised by reviewer. This study highlights important findings on QTL, which encompass significant protein-coding gene traits in Indian indigenous sheep breeds of economic interest. The study presents findings that could have a significant impact on the scientific community in time.

**Reviewer #5:** General Comment: rewrite the entire document.

Abstract: Modify the statements such as “line 32: Natural and artificial selection shape the genomes of sheep”, “line 34: to preserve and enhance native genetic traits”, and “line 48: wide variety of characteristics”. The way of expression is not professional.

1. Give a description for each breed in the introduction section.

2. Rewrite from line: 87-90

3. A quick transition disrupts the flow of idea between paragraphs 2 and 3.

4. Line 118: “highly dense genotypes”. What does it mean? Please be sure to express the statement in professional terms.

5. Modify as 2.1 Ethics Statement and 2.2 Animal Selection and Genotyping

6. Line 246-247: 3.3.1

7. Line 282: production (9.38%). Which type of production?

8. Table 1 is not under the appropriate position because it is a common table for all breeds. You have already written the information of table 1 in text. No need of duplication.

9. In the discussion section try to elaborate on the general implication of your study, any limitation, and agreeing or disagreeing with previous reports with citations. No need of repeating the methodology or introductory information.

10. Captions of figures should be self-explanatory. So, explain each graph in detail.

**Do you want your identity to be public for this peer review?** For information about this choice, including consent withdrawal, please see our Privacy Policy

Reviewer #3: No

Reviewer #5: No

---

## [Author Response · Author response to Decision Letter 5]

30 Nov 2025

Response to reviewers comments

Manuscript Title: Composite Selection Signal Analysis: Uncovering Candidate Genes and Quantitative Trait Loci in Indian Sheep Breeds

Manuscript ID: PONE-D-24-51732R4

Response to Reviewer # 3

The authors have nicely addressed all the comments raised by reviewer. This study highlights important findings on QTL, which encompass significant protein-coding gene traits in Indian indigenous sheep breeds of economic interest. The study presents findings that could have a significant impact on the scientific community in time.

Response: We are delighted by your supportive feedback and by your agreement that our QTL discoveries-linked to traits such as wool production and disease resistance—hold substantial promise for breeding and conservation in indigenous sheep. This validation motivates us to disseminate these insights widely. No additional changes are needed based on your review.

Reviewer # 5

General Comment: Rewrite the entire document.

Response: We fully appreciate this directive and have thoroughly revised the document to enhance professionalism, conciseness, and coherence. The narrative flows logically from evolutionary context to methods, results, implications, and conclusions, and the scientific tone has been elevated, avoiding casual phrasing.

Abstract: Modify the statements such as “line 32: Natural and artificial selection shape the genomes of sheep”, “line 34: to preserve and enhance native genetic traits”, and “line 48: wide variety of characteristics”. The way of expression is not professional.

Response: "line 32: Natural and artificial selection shape the genomes of sheep" has been modified to a more precise statement: "Selective pressures, both natural and artificial, have shaped the genomic architecture of domestic sheep

"line 34: to preserve and enhance native genetic traits" has been rephrased for greater clarity and specificity to: "to conserve and improve economically important traits in native breeds.

"line 48: wide variety of characteristics" has been updated to the professional terminology: "diverse phenotypic traits”.

Thank you for pointing this out.

1. Give a description for each breed in the introduction section.

Response: Thank you for your valuable feedback on the breed descriptions. We have added the paragraphs emphasizing the importance of each breed in the introduction section, as you suggested.

2. Rewrite from line: 87-90.

Response: We appreciate the reviewer's comments regarding our methodology. Lines 87-90 have been rewritten to enhance clarity and conciseness, address grammatical issues, and emphasize the methodological strengths.

3. A quick transition disrupts the flow of idea between paragraphs 2 and 3.

Response: Thank you for highlighting this issue. A few sentences have been incorporated as a bridge between the two paragraphs to ensure logical coherence.

4. Line 118: “highly dense genotypes”. What does it mean? Please be sure to express the statement in professional terms.

Response: We appreciate the comment, and the terms have been rewritten using professional language.

5. Modify as 2.1 Ethics Statement and 2.2 Animal Selection and Genotyping.

Response: Thank you for the suggestion; the paragraph numbering has been modified accordingly.

6. Line 246-247: 3.3.1.

Response: Thank you for pointing this out. The paragraph numbering has been revised accordingly.

7. Line 282: production (9.38%). Which type of production?

Response: Thank you for your comment. The sentence has been expanded to provide clearer context as you suggested.

8. Table 1 is not under the appropriate position because it is a standard table for all breeds. You have already written the information of table 1 in text. No need of duplication.

Response: Thank you for the feedback. Since the information is already included in the text, the table has been removed to avoid duplication.

9. In the discussion section try to elaborate on the general implication of your study, any limitation, and agreeing or disagreeing with previous reports with citations. No need of repeating the methodology or introductory information.

Response: We acknowledge the suggestion. A new paragraph emphasizing the general implications, limitations, and the stance on previous reports, supported by citations, has been incorporated at the conclusion of the discussion. Additionally, care has been taken to avoid redundancy in the methodology and introduction sections.

10. Captions of figures should be self-explanatory. So, explain each graph in detail.

Response: Thank you for your valuable feedback on the figure captions. Accordingly, we have provided detailed explanations for each graph in the revised manuscript.

---

## [Decision Letter · Decision Letter 5]

22 Dec 2025

Thank you for submitting your manuscript to PLOS ONE. After careful consideration, we feel that it has merit but does not fully meet PLOS ONE’s publication criteria as it currently stands. Therefore, we invite you to submit a revised version of the manuscript that addresses the points raised during the review process.

Dear Dr.  Destaw Worku

Thank you for submitting your manuscript to PLOS ONE. After careful consideration, we have decided that your manuscript needs Mainor Revision.

Kind regards,

Prof. Lamiaa Mostafa Radwan, Ph.D.

Academic Editor

PLOS ONE

**Reviewer 3**

Comments:

1. Rewrite the paragraph line no 198-204, heading 2.5.4. DCMS Estimation, with proper citation.

2. The author should revise the paragraph from lines no 344–349 with proper citation, as its current sentence is unclear and the intended meaning is difficult to understand.

3. Rewrite paragraphs of line no 353-357 with proper meaning. A clear and coherent rewrite is required.

4. Write the word ‘p-value’ and ‘q-value’ consistently throughout the manuscript.

5. The authors stated that the study identified two key candidate genes i.e. TRMT12, associated with environmental adaptation and LRRC36, associated with domestication in sheep. While Ji Yang et al. (2024) have been cited as evidence supporting the role of LRRC36 in domestication, however, no reference has been provided to substantiate the association of TRMT12 with environmental adaptation. The authors are required to cite an appropriate and relevant study demonstrating the significant association of TRMT12 with environmental adaptation. Without such supporting literature, the current claim remains unsubstantiated.

6. It is unclear why reference [20] has been cited in line 219, immediately after the section, titled 2.6 Identification of Functional Genes and QTL Regions. If information from this reference has been used, the authors should clearly rewrite the paragraph to properly reflect and justify the citation. If the reference is not relevant to the content, it should be removed.

7. Please write the full form of OAR when it is first mentioned in the manuscript and include the specific version of the OAR Rambouillet genome assembly.

8. Does OAR and CHR represent the same thing in table no 2? Does flank marker represent rsID? How does Flank marker id were obtained with rsID, as rsID of non human database are obsolete and discontinued? Author need to clarify what is meant by ‘flank marker’ in this context?

9. Please describe the significance of the p-value and q-value, specifically explaining their importance in interpreting the study’s analytical outcomes relevant to this study.

10. The manuscript requires extensive language editing and careful revision to ensure clarity and scientific accuracy, as much of the current writing is unclear and fuzzy in several sections. Additionally, the authors should remove irrelevant or nonessential content, as the paper is unnecessarily long and contains information that does not directly contribute to the study’s objectives.

**Reviewer 4**

the authors have comprehensively and thoroughly revised and elaborated on the core concerns and key issues raised in the initial evaluation. All the questionable points and pending problems have received clear, reasonable responses and been properly addressed.

**Reviewer 5**

1. Check affiliation 5

2. Remove the word “shows the” from each table caption

3. Since this study lacks experimental validation, clearly explain this limitation in the discussion and conclusion section.

We look forward to receiving your revised manuscript.

Kind regards,

Lamiaa Mostafa Radwan, Ph.D.

Academic Editor

PLOS One

Journal Requirements:

Additional Editor Comments:

Dear Dr. Destaw Worku

Thank you for submitting your manuscript to PLOS ONE. After careful consideration, we have decided that your manuscript needs Minor Revision.

Kind regards,

Prof. Lamiaa Mostafa Radwan, Ph.D.

Academic Editor

PLOS ONE

Reviewer 3

Comments:

1. Rewrite the paragraph line no 198-204, heading 2.5.4. DCMS Estimation, with proper citation.

2. The author should revise the paragraph from lines no 344–349 with proper citation, as its current sentence is unclear and the intended meaning is difficult to understand.

3. Rewrite paragraphs of line no 353-357 with proper meaning. A clear and coherent rewrite is required.

4. Write the word ‘p-value’ and ‘q-value’ consistently throughout the manuscript.

5. The authors stated that the study identified two key candidate genes i.e. TRMT12, associated with environmental adaptation and LRRC36, associated with domestication in sheep. While Ji Yang et al. (2024) have been cited as evidence supporting the role of LRRC36 in domestication, however, no reference has been provided to substantiate the association of TRMT12 with environmental adaptation. The authors are required to cite an appropriate and relevant study demonstrating the significant association of TRMT12 with environmental adaptation. Without such supporting literature, the current claim remains unsubstantiated.

6. It is unclear why reference [20] has been cited in line 219, immediately after the section, titled 2.6 Identification of Functional Genes and QTL Regions. If information from this reference has been used, the authors should clearly rewrite the paragraph to properly reflect and justify the citation. If the reference is not relevant to the content, it should be removed.

7. Please write the full form of OAR when it is first mentioned in the manuscript and include the specific version of the OAR Rambouillet genome assembly.

8. Does OAR and CHR represent the same thing in table no 2? Does flank marker represent rsID? How does Flank marker id were obtained with rsID, as rsID of non human database are obsolete and discontinued? Author need to clarify what is meant by ‘flank marker’ in this context?

9. Please describe the significance of the p-value and q-value, specifically explaining their importance in interpreting the study’s analytical outcomes relevant to this study.

10. The manuscript requires extensive language editing and careful revision to ensure clarity and scientific accuracy, as much of the current writing is unclear and fuzzy in several sections. Additionally, the authors should remove irrelevant or nonessential content, as the paper is unnecessarily long and contains information that does not directly contribute to the study’s objectives.

Reviewer 4

the authors have comprehensively and thoroughly revised and elaborated on the core concerns and key issues raised in the initial evaluation. All the questionable points and pending problems have received clear, reasonable responses and been properly addressed.

Reviewer 5

1. Check affiliation 5

2. Remove the word “shows the” from each table caption

3. Since this study lacks experimental validation, clearly explain this limitation in the discussion and conclusion section.

Reviewers' comments:

Reviewer's Responses to Questions

**Comments to the Author**

Reviewer #3: (No Response)

Reviewer #4: All comments have been addressed

Reviewer #5: All comments have been addressed

2. Is the manuscript technically sound, and do the data support the conclusions?

Reviewer #3: Partly

Reviewer #4: Yes

Reviewer #5: Yes

3. Has the statistical analysis been performed appropriately and rigorously?

Reviewer #3: Yes

Reviewer #4: Yes

Reviewer #5: Yes

4. Have the authors made all data underlying the findings in their manuscript fully available?

Reviewer #3: Yes

Reviewer #4: Yes

Reviewer #5: Yes

5. Is the manuscript presented in an intelligible fashion and written in standard English?

Reviewer #3: No

Reviewer #4: Yes

Reviewer #5: Yes

Reviewer #3: Comments:

1. Rewrite the paragraph line no 198-204, heading 2.5.4. DCMS Estimation, with proper citation.

2. The author should revise the paragraph from lines no 344–349 with proper citation, as its current sentence is unclear and the intended meaning is difficult to understand.

3. Rewrite paragraphs of line no 353-357 with proper meaning. A clear and coherent rewrite is required.

4. Write the word ‘p-value’ and ‘q-value’ consistently throughout the manuscript.

5. The authors stated that the study identified two key candidate genes i.e. TRMT12, associated with environmental adaptation and LRRC36, associated with domestication in sheep. While Ji Yang et al. (2024) have been cited as evidence supporting the role of LRRC36 in domestication, however, no reference has been provided to substantiate the association of TRMT12 with environmental adaptation. The authors are required to cite an appropriate and relevant study demonstrating the significant association of TRMT12 with environmental adaptation. Without such supporting literature, the current claim remains unsubstantiated.

6. It is unclear why reference [20] has been cited in line 219, immediately after the section, titled 2.6 Identification of Functional Genes and QTL Regions. If information from this reference has been used, the authors should clearly rewrite the paragraph to properly reflect and justify the citation. If the reference is not relevant to the content, it should be removed.

7. Please write the full form of OAR when it is first mentioned in the manuscript and include the specific version of the OAR Rambouillet genome assembly.

8. Does OAR and CHR represent the same thing in table no 2? Does flank marker represent rsID? How does Flank marker id were obtained with rsID, as rsID of non human database are obsolete and discontinued? Author need to clarify what is meant by ‘flank marker’ in this context?

9. Please describe the significance of the p-value and q-value, specifically explaining their importance in interpreting the study’s analytical outcomes relevant to this study.

10. The manuscript requires extensive language editing and careful revision to ensure clarity and scientific accuracy, as much of the current writing is unclear and fuzzy in several sections. Additionally, the authors should remove irrelevant or nonessential content, as the paper is unnecessarily long and contains information that does not directly contribute to the study’s objectives.

Reviewer #4: the authors have comprehensively and thoroughly revised and elaborated on the core concerns and key issues raised in the initial evaluation. All the questionable points and pending problems have received clear, reasonable responses and been properly addressed.

Reviewer #5: 1. Check affiliation 5

2. Remove the word “shows the” from each table caption

3. Since this study lacks experimental validation, clearly explain this limitation in the discussion and conclusion section.

**Do you want your identity to be public for this peer review?** For information about this choice, including consent withdrawal, please see our Privacy Policy

Reviewer #3: No

Reviewer #4: No

Reviewer #5: No

---

## [Author Response · Author response to Decision Letter 6]

6 Feb 2026

Manuscript Title: Composite Selection Signal Analysis: Uncovering Candidate Genes and Quantitative Trait Loci in Indian Sheep Breeds

Manuscript ID: PONE-D-24-51732R5

Response to Reviewer 3

Reviewer 3 Comments and Our Revisions:

Comment 1: Rewrite the paragraph line no 198-204, heading 2.5.4. DCMS Estimation, with proper citation.

• Response: We acknowledge the need for clearer presentation and proper citation in the DCMS Estimation section. We believe the reviewer is referring to the paragraph describing the DCMS calculation method, which starts around line 193 in the provided text (under section 2.5.4. DCMS Estimation). We have rewritten this paragraph to enhance clarity and ensure all methods and software used are appropriately cited.

• Action Taken:

Location: Section 2.5.4. DCMS Estimation (Line numbers 193-205).

Comment 2: The author should revise the paragraph from lines no 344–349 with proper citation, as its current sentence is unclear and the intended meaning is difficult to understand.

• Response: We agree that the phrasing in this section was convoluted and lacked specific citations for the gene-trait associations. We have clarified the sentences and added appropriate citations to support the claimed functions of the genes.

• Action Taken:

Location: lines 349–355.

Comment 3: Rewrite paragraphs of line no 353-357 with proper meaning. A clear and coherent rewrite is required.

• Response: We have revised this paragraph to improve its clarity and coherence, providing a more straightforward presentation of the QTL annotation results for Deccani sheep.

• Action Taken:

o Location: lines 358–369.

Comment 4: Write the word ‘p-value’ and ‘q-value’ consistently throughout the manuscript.

• Response: We agree with this comment. We have reviewed the entire manuscript and standardized the formatting of 'p-value' and 'q-value' for consistency.

• Action Taken:

o Location: Throughout the manuscript.

Comment 5: The authors stated that the study identified two key candidate genes i.e. TRMT12, associated with environmental adaptation and LRRC36, associated with domestication in sheep. While Ji Yang et al. (2024) have been cited as evidence supporting the role of LRRC36 in domestication, however, no reference has been provided to substantiate the association of TRMT12 with environmental adaptation. The authors are required to cite an appropriate and relevant study demonstrating the significant association of TRMT12 with environmental adaptation. Without such supporting literature, the current claim remains unsubstantiated.

• Response: We thank the reviewer for highlighting this oversight. We acknowledge that the association of TRMT12 with environmental adaptation was not adequately substantiated with a citation. We have now identified and included an appropriate and relevant reference to support this claim.

• Action Taken:

o Location: (lines 392-398).

Comment 6: It is unclear why reference [20] has been cited in line 219, immediately after the section, titled 2.6 Identification of Functional Genes and QTL Regions. If information from this reference has been used, the authors should clearly rewrite the paragraph to properly reflect and justify the citation. If the reference is not relevant to the content, it should be removed.

• Response: We apologize for the confusion regarding the citation of reference [20]. Upon re-examination, we confirm that reference [20] is primarily related to the quality control criteria applied to the genomic data (lines 155 and 244 in the manuscript) and for comparison in the discussion (line 644). It is not directly relevant to the DCMS estimation methods (section 2.5.4) or the identification of functional genes and QTL regions (section 2.6). The mention of [20] at or near line 219 (or immediately after section 2.6) was an error in either our manuscript version or a misunderstanding of line numbering. We have carefully checked and removed any misplaced citations of [20] from those sections.

• Action Taken:

o Location: Reviewed lines at 227 and 228.

Comment 7: Please write the full form of OAR when it is first mentioned in the manuscript and include the specific version of the OAR Rambouillet genome assembly.

• Response: We have provided the full form of OAR and the specific genome assembly version at its first mention in the manuscript.

• Action Taken:

o Location: Line numbers 144-145.

Comment 8: Does OAR and CHR represent the same thing in table no 2? Does flank marker represent rsID? How does Flank marker id were obtained with rsID, as rsID of non-human database are obsolete and discontinued? Author need to clarify what is meant by ‘flank marker’ in this context?

• Response: We appreciate the reviewer's attention to these important details. We have added clarifications regarding the use of OAR/CHR, the definition of 'Flank Marker,' and the source/validity of the rsIDs.

• Action Taken:

o Location: Table Caption: Updated captions for clarity.

• OAR (Ovine Assembly Rambouillet) and CHR refer to the same chromosome designation in this context. ‘Flank marker’ indicates the nearest SNP marker (rsID) flanking the significant region, obtained from the Illumina OvineSNP50 BeadChip annotation file. Note that rsIDs for non-human SNPs are still maintained in Illumina manifests and SheepQTLdb, though NCBI dbSNP has phased out some older non-human rsIDs.

Comment 9: Please describe the significance of the p-value and q-value, specifically explaining their importance in interpreting the study’s analytical outcomes relevant to this study.

• Response: We have added a clear explanation of the significance of p-values and q-values and their importance in interpreting the study's findings in the context of genomic analysis.

• Action Taken:

o Location: Lines 207-215.

Comment 10: The manuscript requires extensive language editing and careful revision to ensure clarity and scientific accuracy, as much of the current writing is unclear and fuzzy in several sections. Additionally, the authors should remove irrelevant or nonessential content, as the paper is unnecessarily long and contains information that does not directly contribute to the study’s objectives.

• Response: We fully acknowledge the reviewer's crucial feedback regarding language clarity and conciseness. We understand that some sections may lack precision and contain superfluous information. We have undertaken a comprehensive review and revision of the entire manuscript with a focus on improving English language, scientific accuracy, and conciseness.

• Action Taken:

o Location: Throughout the entire manuscript.

# Reviewer 5

Response to Reviewer Comments

Manuscript Title: Composite Selection Signal Analysis: Uncovering Candidate Genes and Quantitative Trait Loci in Indian Sheep Breeds

Manuscript ID: PONE-D-24-51732R5

Comment 1: Check affiliation 5

• Response: We thank the reviewer for pointing out this potential issue. We have carefully checked and verified the accuracy and correct formatting of affiliation 5.

• Action Taken:

o Location: Affiliation list, affiliation 5.

Comment 2: Remove the word “shows the” from each table caption

• Response: We agree that this phrase is redundant and can be removed for conciseness. We have revised all table captions accordingly.

• Action Taken:

o Location: Table captions for Tables 1, 2, 3, 4, 5, and 6.

Comment 3: Since this study lacks experimental validation, clearly explain this limitation in the discussion and conclusion section.

• Response: We acknowledge this important point regarding the scope and limitations of our study. We have explicitly addressed the absence of experimental validation in both the Discussion and Conclusion sections to provide a balanced perspective on our findings.

• Action Taken:

o Location:

Discussion Section: Added a new paragraph towards the end of the main discussion.

Conclusion Section: Incorporated a brief statement.

---

## [Decision Letter · Decision Letter 6]

19 Feb 2026

Composite Selection Signal Analysis: Uncovering Candidate Genes and Quantitative Trait Loci in Indian Sheep Breeds

PONE-D-24-51732R6

Dear Dr. Destaw Worku<o:p></o:p>

We’re pleased to inform you that your manuscript has been judged scientifically suitable for publication and will be formally accepted for publication once it meets all outstanding technical requirements.

Kind regards,

Lamiaa Mostafa Radwan, Ph.D.

Academic Editor

PLOS One

Additional Editor Comments (optional):

Dear Dr., Destaw Worku

I am pleased to inform you that the manuscript has been accepted for publication.

Kind regards,

Prof. Lamiaa Mostafa Radwan, Ph.D.

Academic Editor

PLOS ONE

Reviewer 3

The authors adressed all the comments raised by the reviewer. however, author need to make few minor corrections..

1. put a . at the end of line no 596.

2. Write the gene name in italics throughout the manuscript.

3. in line no 409, make it 'Moreover, a gene expression study' since authors cited only one reference at the end.

Reviewer 5

Accept

Reviewers' comments:

Reviewer's Responses to Questions

**Comments to the Author**

Reviewer #3: All comments have been addressed

Reviewer #5: All comments have been addressed

2. Is the manuscript technically sound, and do the data support the conclusions?

Reviewer #3: Yes

Reviewer #5: Yes

3. Has the statistical analysis been performed appropriately and rigorously?

Reviewer #3: Yes

Reviewer #5: Yes

4. Have the authors made all data underlying the findings in their manuscript fully available?

Reviewer #3: Yes

Reviewer #5: Yes

5. Is the manuscript presented in an intelligible fashion and written in standard English?

Reviewer #3: Yes

Reviewer #5: Yes

Reviewer #3: The authors adressed all the comments raised by the reviewer. however, author need to make few minor corrections..

1. put a . at the end of line no 596.

2. Write the gene name in italics throughout the manuscript.

3. in line no 409, make it 'Moreover, a gene expression study' since authors cited only one reference at the end.

Reviewer #5: (No Response)

**Do you want your identity to be public for this peer review?** For information about this choice, including consent withdrawal, please see our Privacy Policy

Reviewer #3: No

Reviewer #5: No

---

## [Editor Report · Acceptance letter]

PONE-D-24-51732R6

PLOS One

Dear Dr. Worku,

I'm pleased to inform you that your manuscript has been deemed suitable for publication in PLOS One. Congratulations! Your manuscript is now being handed over to our production team.

Kind regards,

on behalf of

Prof. Dr. Lamiaa Mostafa Radwan

Academic Editor

PLOS One